# FZOO: Fast Zeroth-Order Optimizer for Fine-Tuning Large Language Models towards Adam-Scale Speed

**Sizhe Dang**[*], **Yangyang Guo**[*], **Yanjun Zhao**[*]**& Xiaodong Zheng**
Xi'an Jiaotong University
{darknight1118, yangyangguo, yanjun.zhao, xiaodongzheng}@stu.xjtu.edu.cn

**Guang Dai**
SGIT AI Lab, State Grid Corporation of China
guang.dai@gmail.com

**Ivor Tsang**
CFAR and IHPC, Agency for Science, Technology
and Research (A*STAR)
Ivor_tsang@a-star.edu.sg

**Haishan Ye**[†]
Xi'an Jiaotong University & SGIT AI Lab, State Grid Corporation of China
yehaishan@xjtu.edu.cn

## Abstract

Fine-tuning large language models (LLMs) often faces GPU memory bottlenecks: the backward pass of first-order optimizers like Adam increases memory usage to more than 10 times the inference level (e.g., 633 GB for OPT-30B). Zeroth-order (ZO) optimizers avoid this cost by estimating gradients only from forward passes, yet existing methods like MeZO usually need tens of times more steps to converge. Can this trade-off between speed and memory in ZO be fundamentally improved? Normalized-SGD, for instance, demonstrates strong empirical performance with greater memory efficiency than Adam. In light of this, we introduce **FZOO**, a Fast Zeroth-Order Optimizer towards Adam-Scale Speed. On the one hand, FZOO reduces the total forward passes needed for convergence by employing batched one-sided estimates that adapt step-sizes based on the standard deviation of batch losses. On the other hand, it accelerates per-batch computation through the use of Rademacher random vector (±1) perturbations, which also enables further speedups through batched evaluation. Extensive experiments on diverse models (including RoBERTa-large, the OPT family (350M-66B), Phi-2, and Llama3) across 11 varied downstream tasks validate FZOO's effectiveness. On average, FZOO outperforms MeZO by +3% in accuracy while requiring $3\times$ fewer forward passes. Notably, for the RoBERTa-large model, FZOO achieves average improvements of +5.6% in accuracy and $18\times$ reduction in forward passes compared to MeZO, achieving convergence speeds comparable to Adam. We also provide theoretical analysis proving FZOO's formal equivalence to a normalized-SGD update rule and establishing its convergence guarantees. Beyond full-parameter tuning, FZOO plugs smoothly into PEFT techniques, unlocking even larger memory savings. Taken together, our results make single-GPU, high-speed, full-parameter fine-tuning realistic today and point toward future work on memory-efficient pre-training. **Code**: https://github.com/DKmiyan/FZOO

## 1 Introduction

Large language models (LLMs) have become the workhorse of natural-language and multi-modal applications (Devlin et al., 2019; Gururangan et al., 2020; Ouyang et al., 2022), yet they often require

---

[*]Equal contribution. This work was completed at SGIT AI Lab, State Grid Corporation of China.
[†]Corresponding author.

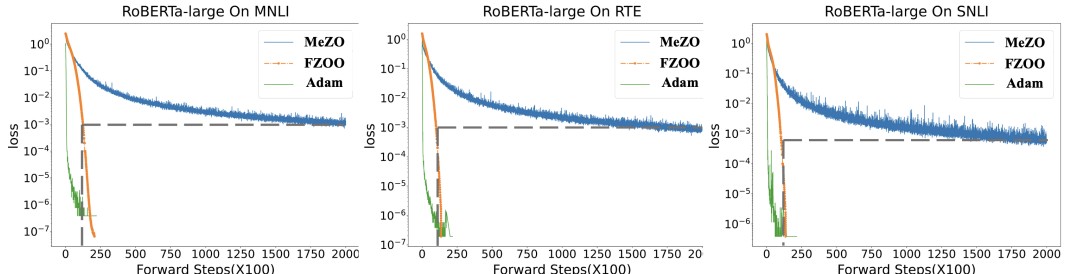

Figure 1: Performance of MeZO, Adam and FZOO on different tasks when fine-tuning RoBERTa-large model. For the sake of uniform comparison, we convert Adam's backward pass into 3 forward passes. FZOO is 18× faster than MeZO in forward steps, and Adam is 20× faster than MeZO.

additional fine-tuning to excel on downstream tasks. Current practice relies on first-order optimizers such as Adam (Kingma & Ba, 2015), whose backward pass stores all activations and gradients, linearly multiplying the memory cost—fine-tuning an OPT-30B model then consumes **633 GB** of GPU memory, more than **10×** its inference memory cost. This creates the *memory wall* that now limits model training in resource-constrained scenarios.

Two main strategies have been explored to overcome this wall. *Parameter-efficient tuning* (PEFT) (Hu et al., 2022; Li & Liang, 2021; Dettmers et al., 2023; Zhao et al., 2024; Pan et al., 2024) (e.g., LoRA (Hu et al., 2022)) updates only a small subset of weights and thus lowers memory cost, but it still requires the backward pass and can lag full-parameter tuning (FT) on some difficult tasks. The other approach turns to *zeroth-order* (ZO) optimization (Malladi et al., 2023; Zhao et al., 2025c): methods such as MeZO (Malladi et al., 2023) use multiple forward passes to replace the backward pass when estimating gradients. This brings the memory overhead of fine-tuning down to the inference level, but this benefit comes with a significant trade-off in speed, as convergence is often about 20× slower than Adam on RoBERTa-large (Figure 1). **Does the use of zeroth-order methods always lead to a trade-off between speed and memory?**

We believe the answer is **no**. This is because ZO performance can be substantially enhanced in three key areas: (i) **Adaptive Step-Sizes.** While Adam uses costly momentum for adaptive steps, normalized-SGD adapts efficiently via gradient normalization without this overhead. Existing ZO methods, however, use inefficient fixed step-sizes for LLM fine-tuning. (ii) **Efficient perturbation.** Furthermore, each ZO gradient estimate typically requires multiple forward queries. However, the overall speed of this multi-query process is limited by the cost of conventional perturbation sampling (e.g., Gaussian noise); because many perturbations must be generated, this compute does not diminish even with batched execution, and can not utilize hardware parallelism. (iii) **Engineering Optimization.** Theoretically, one Adam step is twice as fast as a MeZO step (Adam's backward pass takes roughly three times longer than a forward pass (Alman & Song, 2024), making Adam equivalent to four forward passes versus MeZO's two). Yet, due to insufficient forward-pass optimizations, MeZO's actual per-step execution often exceeds Adam's. Taken together, these observations suggest that a well-engineered ZO optimizer could achieve **Order-of-magnitude Adam-like convergence speed with inference-level memory cost**.

To harness these opportunities for ZO enhancement, we introduce **FZOO**, a **F**ast **Z**eroth-**O**rder **O**ptimizer. FZOO achieves its significant speed-up through two primary strategies: First, it reduces total convergence passes by employing **batched one-sided estimates**, where the standard deviation of batch losses informs an **adaptive step-size rule**. Second, it accelerates each batch computation using efficient **Rademacher random-vector perturbations** (±1), **enabling** batched parallelism for query evaluations. Specifically, our contributions are:

1. **FZOO**, a novel ZO optimizer that uniquely adapts principles from normalized-SGD to the ZO regime. It is designed to **move towards Adam-scale** convergence speed while operating at an **inference-level memory usage**.

2. We provide rigorous theoretical backing for FZOO, including a formal proof of its equivalence to a normalized-SGD update rule and a comprehensive analysis establishing its **convergence guarantees** under standard conditions.

3. Extensive FT experiments on diverse models (RoBERTa-large, OPT family 350M–66B, Phi-2, and Llama3) across 11 downstream tasks demonstrate FZOO's superior efficiency. On average, FZOO surpasses the MeZO baseline by **+3%** in accuracy with **3×** fewer forward passes. Notably, as show in Fig. 1 for the RoBERTa-large model it achieves average improvements of **+5.6%** in accuracy and an **18×** reduction in forward passes compared to MeZO, reaching effective convergence speeds **comparable to Adam**.

4. FZOO exhibits strong practical utility, smoothly integrating with PEFT techniques (Dettmers et al., 2022a;b) to unlock **even larger memory savings**. Furthermore, it effectively optimizes both differentiable and non-differentiable objectives during FT.

## 2 RELATED WORKS

We provide a brief overview of first-order, zeroth-order and batch-based zeroth-order optimizers for large language models; see Appendix A for details.

**First-order adaptive methods** Standard LLM optimizers include SGD and its refinements (Duchi et al., 2011) like Adam (Kingma & Ba, 2015), which uses memory-intensive momentum for its adaptive step-sizes. AdamW (Loshchilov & Hutter, 2019) improves generalization by refining weight decay. Inspiringly, methods like normalized-SGD (Bernstein et al., 2018) also adapt steps effectively, often rivaling Adam's performance but crucially without this significant momentum overhead, offering a key insight for our work.

**Zeroth-order optimizers** ZO methods sidestep back-prop by estimating gradients with only forward passes. The classic SPSA estimator (Spall, 1992) has powered applications from distributed control (Tang et al., 2021; Hajinezhad & Zavlanos, 2018) to black-box adversarial attacks (Chen et al., 2017; Cai et al., 2021; Liu et al., 2019a; Ye et al., 2019). MeZO (Malladi et al., 2023) first showed that such forward-only updates can fine-tune LLMs to high accuracy while cutting GPU memory by several times (up to 12×). HiZOO (Zhao et al., 2025c) augments MeZO with diagonal Hessian cues to accelerate convergence.A recent benchmark (Zhang et al., 2024) catalogues many other ZO estimators.

**Batch-based zeroth-order optimizers** A batch-oriented refinement of ZO methods estimates gradients by evaluating a batch of forward step. Recent work like ReLIZO (Wang et al., 2024) reuses or correlates directions to boost sample efficiency. Augmented Random Search (Mania et al., 2018) and Evolution Strategies (Sun et al., 2022a;b) adopt similar population-based updates. Notably, DeepZero (Chen et al., 2023) proposes a framework to scale up ZO optimization via coordinate-wise parallelization and feature reuse, which shares the similar philosophy of activation recycling with our work.

## 3 METHODS

In the following, we first briefly review the classical zeroth-order gradient estimator employed in MeZO (Section 3.1). We then present the complete workflow of our **FZOO** optimizer (Section 3.2), shown in Figure 2, introduce a parallel perturbation strategy that accelerates computation (Section 3.3), and finally offer a theoretical analysis of the proposed method (Section 3.4, 3.5).

### 3.1 PRELIMINARIES

Consider a labelled dataset $\mathcal{D} = \{(x_i, y_i)\}_{i=1}^{|\mathcal{D}|}$ and a mini-batch $\mathcal{B} \subset \mathcal{D}$ of size $B$. Let $\theta \in \mathbb{R}^d$ denote all trainable parameters of the LLM, and let $L(\theta; \mathcal{B})$ be the empirical loss on $\mathcal{B}$.

**Definition 3.1** (Classical ZO gradient estimation). Given a perturbation radius $\epsilon > 0$ and $z \in \mathbb{R}^d$ sampled as $z \sim \mathcal{N}(0, I_d)$, Classical ZO estimates the gradient on $\mathcal{B}$ via

$$\hat{\nabla} L(\theta; \mathcal{B}) \; = \; \frac{L(\theta + \epsilon z; \mathcal{B}) - L(\theta - \epsilon z; \mathcal{B})}{2\epsilon} \, z \; \approx \; z z^\top \, \nabla L(\theta; \mathcal{B}). \qquad (1)$$

Averaging equation 1 over $N$ i.i.d. draws $\{z_i\}_{i=1}^n$ yields the $n$-ZO estimator $\hat{\nabla}_N L = \frac{1}{N} \sum_{i=1}^N \hat{\nabla}_i L$.

**Computational cost.** A single ZO estimate needs exactly *two forward passes* and does not store activations for backpropagation. MeZO keeps memory usage at the inference level by passing random

---

**Algorithm 1** FZOO

---

**Require:** parameters $\theta \in \mathbb{R}^d$, loss $L : \mathbb{R}^d \to \mathbb{R}$, step budget $T$, perturbation scale $\epsilon$, batch size $N$, learning rate schedule $\{\eta_t\}$
1: **for** $t = 1, ..., T$ **do**
2:     $\ell, \theta, seeds \leftarrow$ BatchPerturbParameters$(\theta, \epsilon, N)$
3:     $std \leftarrow$ standard deviation of $\ell$
4:     $projected\_grad \leftarrow (\ell - \mathcal{L}(\theta; \mathcal{B}))/(N * std)$
5:     BatchUpdateParameter$(projected\_grad, seeds, \theta, \eta_t)$
6: **end for**
7: **function** BATCHPERTURBPARAMETER$(\theta, \epsilon, N)$
8:     Sample batch $\mathcal{B} \subset \mathcal{D}$; obtain input $X$ and first layer weights $W^{(1)}$
9:     Initialize random seeds $seeds \leftarrow \{s_1, \ldots, s_N\}$
10:     Generate perturbation vectors $u$ using $seeds$
11:     Compute unperturbed activations $F^{(1)} \leftarrow W^{(1)} X$
12:     Compute perturbed activations: $Y_i^{(1)} \leftarrow F^{(1)} + \epsilon(u_i X)$, $i = 1, \ldots, N$
13:     Concatenate activations $Y^{(1)} \leftarrow [Y_1^{(1)}; \ldots; Y_N^{(1)}]$
14:     **for** $j = 2, 3, \ldots$ **do**
15:         $F^{(j)} \leftarrow W^{(j)} Y^{(j-1)}$                   ▷ Compute unperturbed activations at layer $j$
16:         $P^{(j)} \leftarrow \epsilon(u Y^{(j-1)})$                ▷ Compute perturbations at layer $j$
17:         $Y^{(j)} \leftarrow F^{(j)} + P^{(j)}$         ▷ Compute perturbed activations in parallel at layer $j$
18:     **end for**
19:     $\ell \leftarrow \mathcal{L}(Y^{(final)}; \mathcal{B})$                          ▷ Compute final losses in parallel
20:     **return** $\ell, \theta, seeds$
21: **end function**
22: **function** BATCHUPDATEPARAMETER$(projected\_grad, seeds, \theta, \eta)$
23:     **for** $idx, s \in$ enumerate$(seeds)$ **do**
24:         Reset random number generator with seed $s$              ▷ For sampling $u$
25:         **for** $\theta_i \in \theta$ **do**
26:             $u \sim$ Uniform$(\{+1, -1\})$
27:             $\theta_i \leftarrow \theta_i - \eta * projected\_grad_{idx} * u$
28:         **end for**
29:     **end for**
30: **end function**

---

$seeds$ instead of full perturbation $z$. MeZO further observes that choosing $N > 1$ improves statistical stability but increases computation by a factor of $N$; therefore it fixes $N = 1$.

**From Classical ZO to ZO-SGD.** Replacing the back-propagation gradient in SGD with the Classical ZO estimate gives the zeroth-order update $\theta_{t+1} = \theta_t - \eta_t \hat{\nabla} L(\theta_t; \mathcal{B}_t)$. MeZO realises this update in-place with the memory tricks above and serves as the baseline for our forthcoming improvements.

### 3.2 FAST ZEROTH-ORDER OPTIMIZATION

MeZO employs a fixed learning rate. Instead, FZOO dynamically tunes its step size using the standard deviation of loss values gathered from multiple forward passes within each mini-batch. For the enhanced computational efficiency, FZOO also incorporates a one-sided gradient estimation strategy and **we first utilize Rademacher random vectors for perturbations**. Letting $u_1, \ldots, u_N$ be $N$ i.i.d Rademacher random vectors ($u$ has the same shape as each layer's weight $W^{(j)}$), we construct our gradient estimation by function value queries

$$l_i = L(\theta_t + \epsilon u_i; \mathcal{B}_t), \quad \text{and} \quad l_0 = L(\theta_t; \mathcal{B}_t).$$

That is, the gradient estimate $g_t$ is then computed by averaging $N$ one-sided difference estimates

$$g_t = \frac{1}{\epsilon N} \sum_{i=1}^{N} (l_i - l_0) u_i. \tag{2}$$

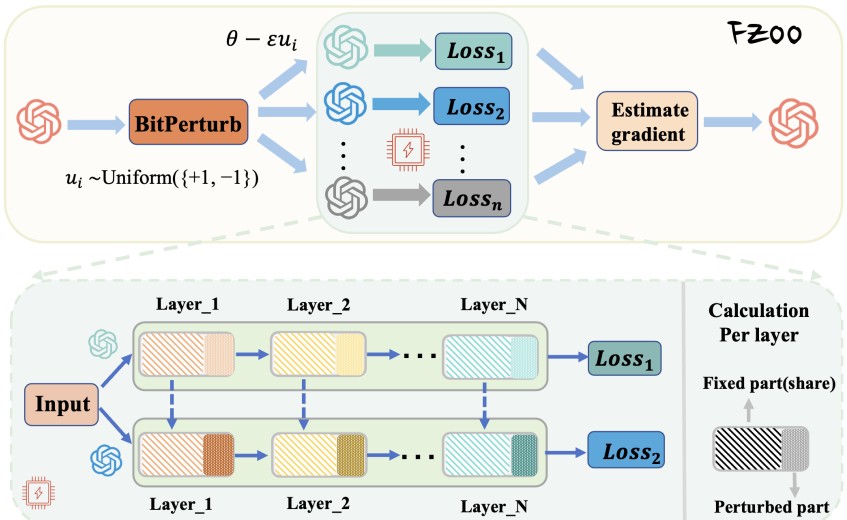

Figure 2: Structure of the of FZOO. The bottom half depicts the toy example of the efficient implementation of batched forward passes.

The estimated variance $\sigma_t^2$ is computed as:

$$\sigma_t^2 = \frac{1}{N-1} \sum_{i=1}^{N} \left( l_i - \frac{1}{N} \sum_{j=1}^{N} l_j \right)^2. \tag{3}$$

Our FZOO updates the parameters according to equation 2 and equation 3:

$$\theta_{t+1} = \theta_t - \eta_t \frac{g_t}{\sigma_t}, \tag{4}$$

where $\eta_t$ is the step size. The detailed implementation of FZOO is listed in Algorithm 1.

The equation 4 shows that FZOO receives larger steps at flat regions (where $\sigma_t$ is small) and smaller steps at steep regions, mirroring Adam-style adaptivity while keeping memory at the inference level.

### 3.2.1 MOVTIVATION OF FZOO

Adaptive first-order methods often estimate local curvature to scale updates. Similar adaptivity can be achieved by methods like normalized-SGD (Bernstein et al., 2018), which adjusts step sizes by normalizing the gradient, making it more memory-efficient compared to Adam. The parameter update follows normalized-SGD:

$$\theta_{t+1} = \theta_t - \eta_t \frac{g_t}{||g_t||}, \tag{5}$$

FZOO is inspired by normalized-SGD. Proposition 3.2 shows that $\sigma_t^2 = |g_t|^2 \cdot \epsilon^2 \cdot \frac{N-1}{N}$ which implies that FZOO is an extension of normalized-SGD to the ZO domain.

**Variants.** *FZOO-R* re-uses half of the losses from the previous mini-batch: $\sigma_t = \mathrm{Std}\big(\{l_i^{\mathrm{curr}}\} \cup \{l_i^{\mathrm{prev}}\}\big)$, achieving a full-batch variance estimate with only half the forward evaluations.

### 3.3 EFFICIENT IMPLEMENTATION OF BATCHED FORWARD PASS

The original ZO pipeline applies a perturbation and then runs a separate forward pass, which cannot take advantage of batch parallel computing. For layer $j$ we split the computation into an unperturbed part $F^{(j)}$ and a perturbation part $P^{(j)}$. Using Gaussian noise forces two full matrix–vector products, so batching recovers little speed-up. Replacing the noise with a Rademacher vector($\pm 1$) $u_i$ changes only the sign bits: $P^{(j)}$ is produced by a bit-level sign flip that degenerates into a single add / subtract, so the kernel issues additions instead of a second matrix–vector multiply, making it markedly faster because addition is cheaper than multiplication (Williams et al., 2009; Chen et al., 2020).

Let $X$ be the input and let $W^{(j)}$ denote the weight matrix of layer $j$. For the **first layer** we compute

$$F^{(1)} = W^{(1)}X, \qquad Y_i^{(1)} = F^{(1)} + \epsilon\,(u_i X), \ i = 1, \dots, N,$$

and we stack the perturbed outputs along the batch dimension: $Y^{(1)} = [Y_1^{(1)}; \dots; Y_N^{(1)}]$.

Let $Y^{(j-1)}$ be the concatenated activations from the previous layer and $U = \operatorname{diag}(u_1, \dots, u_N)$ the block-diagonal sign matrix that broadcasts the $u_i$ across the batch axis. For **subsequent layers** ($j \geq 2$) we compute

$$F^{(j)} = W^{(j)} Y^{(j-1)}, \qquad P^{(j)} = \epsilon\,(U Y^{(j-1)}), \qquad Y^{(j)} = F^{(j)} + P^{(j)}.$$

By executing the $N$ per-layer matrix multiplications in parallel, the wall-clock time is reduced by a factor $p$ compared with sequential evaluation. Combining advantage of parallel computing ($p$), Rademacher vector perturbations ($r$), and the one-sided estimator's halved forward count ($f$) yields the overall speed-up

$$\boxed{f \times \min(p, r)}.$$

On OPT-125M with $N = 8$, our batched scheme($\min(p, r)$) delivers a $1.92\times$ speed-up over the "8 perturbations + 8 forward passes" baseline.

## 3.4 Equivalence to normalized-SGD

**Proposition 3.2.** *Let the stochastic gradient estimation $g_t$ defined in equation 2 and the variance $\sigma_t$ defined in equation 3. Then it holds that*

$$\mathbb{E}\left[\|g_t\|^2\right] = \frac{N + d - 1}{N} \|\nabla L(\theta_t, \mathcal{B}_t)\|^2 + \gamma_t \quad \text{and} \quad \gamma_t = O(\epsilon) \tag{6}$$

$$\mathbb{E}\left[\sigma_t^2\right] = \epsilon^2 \cdot \|\nabla L(\theta_t, \mathcal{B}_t)\|^2 + \zeta_t \quad \text{with} \quad \zeta_t = O(\epsilon^3). \tag{7}$$

*Remark* 3.3. Comparing equation 7 and equation 6, and ignoring the perturbation terms $\zeta_t$ and $\gamma_t$ (which are higher-order terms with respect to $\epsilon$), we can obtain that:

$$\mathbb{E}\left[\|g_t\|^2\right] = \frac{N + d - 1}{N} \cdot \epsilon^{-2} \cdot \mathbb{E}\left[\sigma_t^2\right]$$

Since $\frac{N+d-1}{N} \cdot \epsilon^{-2}$ is a constant independent of the iteration number, $\frac{g_t}{\sigma_t}$ can be regarded as a kind of normalized stochastic gradient, scaled by a constant factor. This theoretically establishes the connection between FZOO and normalized-SGD.

## 3.5 Convergence Analysis

**Assumption 3.4.** Suppose that the loss function $L(\theta)$ is $\mathcal{L}$-smooth, that is, for any $\theta_1, \theta_2 \in \mathbb{R}^d$, it holds that $L(\theta_2) \leq L(\theta_1) + \langle \nabla L(\theta_1), \theta_2 - \theta_1 \rangle + \frac{\mathcal{L}}{2} \|\theta_2 - \theta_1\|^2$.

**Assumption 3.5** (Bounded Variance). The stochastic gradient $\nabla L(\theta, \mathcal{B})$ has bounded variance:

$$\mathbb{E}\left[\|\nabla L(\theta, \mathcal{B})\|^2\right] \leq \|\nabla L(\theta)\|^2 + \mathcal{V}^2,$$

where $\mathcal{V}^2$ is a constant. The above is a standard assumption for stochastic gradient descent.

**Theorem 3.6** (Convergence of FZOO). *Let the objective function $L(\theta)$ be $\mathcal{L}$-smooth, and Assumption 3.5 hold. The update rule is $\theta_{t+1} = \theta_t - \eta_t g_t$, where the effective step size $\eta_t = \frac{\eta}{\sigma_t}$ satisfies $\eta_t \leq \frac{N}{16d\mathcal{L}}$, and the learning rate factor $\eta$ is set as $\eta = \frac{1}{\sqrt{T}}$. Then, after $T$ iterations, the FZOO algorithm satisfies:*

$$\frac{1}{T}\sum_{t=1}^{T} \mathbb{E}\left[\|\nabla L(\theta_t)\|^2\right] \leq \frac{4\sigma_*}{\sqrt{T}} \left( L(\theta_1) - L(\theta^*) + \frac{1}{T}\sum_{t=1}^{T} \left( \frac{4d\mathcal{L}\sigma_t^{-2}\mathcal{V}^2}{N} + \frac{d^3\mathcal{L}^3\sigma_t^{-2}\epsilon^2}{8N} + \frac{Nd\mathcal{L}^3\sigma_t^{-2}\epsilon^2}{2} \right) \right)$$

*where $\theta^*$ denotes the optimal solution (minimum) of $L(\theta)$, and $\sigma_* = \max_{t \in \{1, \dots, T\}} \sigma_t$.*

This theorem shows that the FZOO algorithm converges to a stationary point whose gradient norm close to zero. Especially, our theorem shows that to find obtain $\frac{1}{T}\sum_{t=1}^{T} \mathbb{E}\left[\|\nabla L(\theta_t)\|^2\right] \leq \varepsilon$ with $\varepsilon > 0$, it requires $T = O(\varepsilon^{-2})$ which matches the convergence rate of stochastic gradient descent for nonconvex optimization.

Table 1: Experiments on RoBERTa-large (350M parameters, k=16). All reported numbers are averaged accuracy across 5 runs. In notations like $(2 \times M)$, M represents the memory overhead required for inference. All methods are evaluated under a fixed forward-step budget aligned with MeZO: 200k steps for RoBERTa-large and 40k steps for other models.

| Task Type | SST-2 | SST-5 | SNLI | MNLI | RTE | TREC | Average |
|---|---|---|---|---|---|---|---|
| | —— sentiment —— | | —— natural language inference —— | | | — topic — | |
| Zero-shot | 79.0 | 35.5 | 50.2 | 48.8 | 51.4 | 32.0 | 49.5 |
| LP | 76.0 | 40.3 | 66.0 | 56.5 | 59.4 | 51.3 | 58.3 |
| HiZOO $(2 \times M)$ | 93.2 | 46.2 | 74.6 | 64.9 | 66.8 | 79.8 | 70.9 |
| ZO-Adam $(3 \times M)$ | 90.3 | 38.8 | 69.1 | 59.2 | 61.4 | 52.4 | 61.7 |
| FT $(6 \times M)$ | 91.9 | 47.5 | 77.5 | 70.0 | 66.4 | 85.0 | 74.9 |
| MeZO(prefix) | 90.8 | 45.8 | **71.6** | 63.4 | **65.4** | 80.3 | 69.6 |
| HiZOO(prefix) | 92.3 | **47.2** | 68.8 | 61.6 | **65.4** | 82.0 | 69.6 |
| FZOO(prefix) | **92.9** | 43.7 | 70.8 | **67.0** | 64.6 | **82.8** | **70.3** |
| MeZO | 90.5 | 45.5 | 68.5 | 58.7 | 64.0 | 76.9 | 67.4 |
| FZOO | **93.3** | **47.6** | **75.9** | **64.9** | **67.9** | 78.8 | **71.4** |

# 4 EXPERIMENTS

Large language models (LLMs) fall into two architectural families. Encoder–decoder models—BERT (Devlin et al., 2019), ALBERT (Lan et al., 2020), etc.—tackle language understanding and are trained with a masked-token objective (Section 4.1). Decoder-only models—GPT (Radford et al., 2019; Brown et al., 2020), OPT (Zhang et al., 2022), LLaMA (Touvron et al., 2023), Phi (Li et al., 2023; Gunasekar et al., 2023), and others—focus on text generation and are trained autoregressively (Section 4.2). We benchmark FZOO under full-parameter fine-tuning on both model families, covering differentiable and non-differentiable tasks (Section 4.3), report its memory and wall-clock speed (Section 4.4), and compare it with standard ZO baselines (Section 4.5). Section 4.6 further shows that FZOO is complementary to parameter-efficient methods such as prefix-tuning (Li & Liang, 2021). For a fair comparison with prior ZO baselines that do not employ batched perturbations, our main experiments use the non-parallel variant in Algorithm 3, and speed is reported as the number of forward passes.

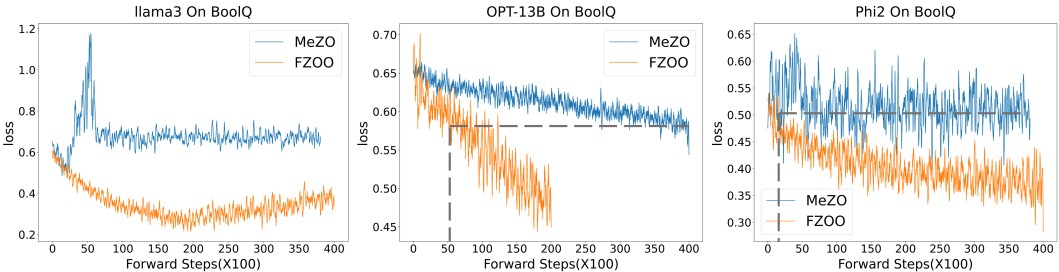

Figure 3: Training loss curves when using MeZO and FZOO to fine-tune different LLMs.

## 4.1 MASKED LANGUAGE MODELS

We perform experiments on RoBERTa-large (350M) (Liu et al., 2019b) across three NLP task paradigms: sentence classification, multiple choice and text generation. Following the experimental setup in (Malladi et al., 2023), we investigate both few-shot and many-shot settings, sampling $k$ examples per class with $k = 16$ (results in Table 1) and $k = 512$ (results in Appendix D.1).

According to prior analysis, the cost of a single backpropagation is roughly three times that of a forward pass (Alman & Song, 2024), making each Adam update equivalent to four forward propagations. As shown in Figure 1, FZOO reduces loss significantly faster than MeZO, achieving an acceleration ratio of $18\times$. With an additional $1.92\times$ parallel speedup, FZOO can potentially surpass Adam in wall-clock efficiency. Moreover, its convergence rate closely approaches that of Adam, underscoring FZOO's strong training efficiency.

Table 1 shows that when optimizing the RoBERTa-large model with MeZO and FZOO, the performance of FZOO is on average **5.9%** higher than that of MeZO on multiple datasets across different NLP tasks. Specifically, FZOO outperforms MeZO more than **10.7%** in both the SNLI and MNLI dataset, showing comparable performance to HiZOO.

Table 2: Experiments on three different models (with 1000 examples) : Classification (SST-2, RTE, CB, BoolQ, WSC, WIC, MultiRC); Multiple Choice (COPA, ReCoRD); Generation (SQuAD, DROP). We highlight the best results between MeZO, HiZOO-L and FZOO in bold to facilitate comparison.

| Model | Method | SST-2 | RTE | CB | BoolQ | WSC | WIC | MultiRC | COPA | ReCoRD | SQuAD | DROP | Average |
|---|---|---|---|---|---|---|---|---|---|---|---|---|---|
| Phi-2 | Adam | 84.4 | 61.6 | 79.0 | 68.8 | 59.6 | 67.7 | 77.8 | 84.0 | 68.7 | 90.4 | 41.1 | 71.2 |
| Phi-2 | MeZO | 86.6 | 67.1 | 75.0 | 72.4 | 59.6 | 54.4 | 78.2 | **86.0** | 71.7 | 85.7 | **37.8** | 70.7 |
| Phi-2 | HiZOO-L | **88.9** | 68.9 | 75.2 | 72.0 | 62.4 | 59.2 | 79.2 | **86.0** | **72.1** | 85.7 | 36.2 | 71.4 |
| Phi-2 | FZOO | 87.4 | **70.4** | **83.9** | **79.3** | 61.5 | 56.7 | **81.3** | **86.0** | 72.0 | **86.7** | 37.4 | **73.0** |
| Llama3 | Adam | 94.6 | 80.7 | 94.6 | 83.3 | 64.4 | 71.6 | 84.7 | 89.0 | 86.9 | 89.7 | 58.4 | 81.6 |
| Llama3 | MeZO | 92.2 | 74.4 | 69.6 | 76.7 | 63.5 | 57.8 | 77.6 | 88.0 | **85.6** | 86.7 | **57.1** | 75.4 |
| Llama3 | HiZOO-L | **94.3** | 75.1 | 69.6 | 77.1 | 63.5 | 57.7 | 77.9 | **89.0** | **85.6** | 87.7 | 49.4 | 75.2 |
| Llama3 | FZOO | **94.3** | **77.6** | 69.6 | **81.8** | 65.4 | 60.8 | **81.5** | 88.0 | 85.3 | **87.9** | 56.5 | **77.2** |
| OPT-13B | Adam | 92.1 | 79.1 | 71.4 | 77.0 | 63.5 | 69.6 | 76.2 | 88.0 | 81.0 | 84.5 | 31.3 | 74.0 |
| OPT-13B | MeZO | 91.4 | 66.1 | 66.0 | 67.6 | 63.5 | 59.4 | 57.3 | 88.0 | **81.7** | 84.7 | **30.9** | 68.8 |
| OPT-13B | HiZOO-L | 92.1 | 68.2 | 67.9 | 66.6 | **65.4** | 59.4 | 61.1 | **89.0** | 81.1 | 63.6 | 22.7 | 67.0 |
| OPT-13B | FZOO | **93.7** | **71.1** | 69.6 | 72.2 | 63.5 | **60.5** | 66.0 | 87.0 | 81.0 | **84.8** | 28.7 | **70.7** |

## 4.2 AUTO-REGRESSIVE LANGUAGE MODELS

Next, we expand the scope of our experiments by testing the Phi-2(2.7B), Llama3(8B) and OPT(13B) models on the same NLP tasks. The results of the experiment in Table 2 show that FZOO outperforms MeZO in most cases, with an average accuracy increase of **2.75%**. Aside from the ReCoRD and DROP datasets, FZOO surpasses MeZO in nearly all other cases.

**FZOO significantly accelerates convergence speed during full-parameter tuning.** As shown in Figure 3, FZOO not only achieves **8×** speedup over MeZO on average in the context of full parameter tuning, but also has lower training loss. In addition, FZOO ultimately achieved a higher absolute accuracy improvement of **7.4%** than MeZO on the BoolQ task.

**FZOO is able to adapt to large models with up to 66B parameters and maintains excellent performance.** As shown in Table 3, on the OPT-66B model, FZOO outperforms MeZO with up to **13.2%** increase and **2.43%** increase on average.

## 4.3 TRAINING WITH NON-DIFFERENTIABLE OBJECTIVES

Our proposed FZOO estimates the gradient by performing multiple forward propagation processes, so the objective function can be a non-differentiable function. Based on the MeZO (Malladi et al., 2023) setup, we conduct some experiments using F1 as the optimization objective. As shown in Table 4, FZOO performs consistently well across OPT models of different scales, surpassing both MeZO and HiZOO-L, and achieving an average F1 improvement of **5.53%** over MeZO.

Table 3: Experiments on the OPT-30B and OPT-66B (both of them use FT).

| Task | SST-2 | RTE | WSC | WIC | Average |
|---|---|---|---|---|---|
| 30B MeZO | 90.6 | 66.4 | **63.5** | 56.3 | 69.2 |
| 30B HiZOO-L | 90.3 | 69.3 | **63.5** | 53.4 | 69.3 |
| 30B FZOO | **91.2** | 69.3 | 63.5 | 60.2 | **71.1** |
| 66B MeZO | 91.2 | 65.7 | **63.5** | 58.9 | 69.8 |
| 66B HiZOO-L | 88.9 | 66.4 | 59.6 | 58.6 | 68.4 |
| 66B FZOO | **93.6** | **74.4** | 58.6 | 59.2 | **71.5** |

Table 4: Non-differentiable optimization objectives (F1): Performance on the SQuAD.

| Model | OPT | | | | | |
|---|---|---|---|---|---|---|
| Size | 125M | 1.3B | 2.7B | 6.7B | 13B | Average |
| Zero-shot | 9.7 | 27.2 | 29.9 | 36.6 | 46.2 | 29.9 |
| MeZO | 44.1 | 72.2 | 77.6 | **80.1** | 78.5 | 70.5 |
| HiZOO-L | 37.6 | 71.9 | 77.5 | 79.5 | 81.8 | 69.7 |
| FZOO | **51.0** | **74.6** | **81.8** | 79.8 | **84.7** | **74.4** |

## 4.4 MEMORY USAGE AND TIME EFFICIENCY ANALYSIS

**Memory Usage** As shown in Figure 4, FZOO and MeZO have the same inference-level memory and do not need to store additional content like HiZOO. Even if Adam is trained with PEFT like prefix, it still requires several times more memory than FZOO.

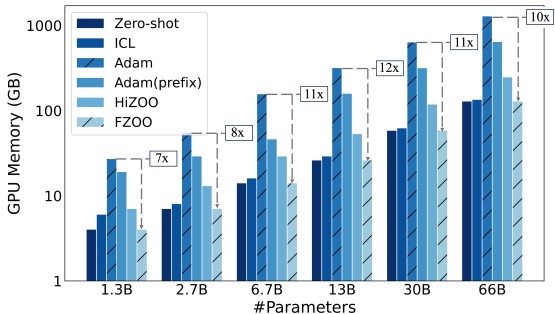

Figure 4: GPU memory consumption with different OPT models and tuning methods on MultiRC (400 tokens per example on average). More details can be found in Appendix E.

Table 5: Wallclock time per step between Adam, MeZO and FZOO(N=8). See Appendix F for more.

| Method | OPT-125M | RoBERTa-large | OPT-1.3B |
|--------|----------|---------------|----------|
| Adam   | 0.1982s  | 0.3930s       | 0.5814s  |
| MeZO   | 0.1368s  | 0.4305s       | 0.7218s  |
| FZOO   | 0.3835s  | 0.6052s       | 1.6628s  |

Table 6: Actual and potential speedup of FZOO for different tasks. Adam commonly achieves 20x speedup than MeZO.

| Task      | SNLI | COPA | WIC  | CB  |
|-----------|------|------|------|-----|
| FZOO      | 20×  | 10×  | 9×   | 8×  |
| Potential | 40×  | 20×  | 18×  | 16× |

**Time Efficiency**  Adaptive step-size scheduling delivers up to $20\times$ speed-up in total steps over MeZO (Table 6). One FZOO step bundles 9 forward passes—$4.5\times$ more than MeZO—yet consumes only $3\times$ the wall-clock time (Appendix F), highlighting the gain from our parallel implementation.

## 4.5 COMPARISON WITH ZO VARIANTS AND HYBRID FRAMEWORKS

We also compare FZOO with a wider range of ZO optimization methods (Zhang et al., 2024). As shown in Table 7, FZOO consistently outperforms all baselines. Compared to ZO-Adam, our FZOO achieves an average accuracy improvement of **5.77%** under full-parameter tuning, and **4.21%** under prefix-tuning, while using **40.5%** of the GPU memory.

Table 7: Performance comparison on SST2(Robert-Large and OPT-1.3B) and COPA(OPT-13B) using different ZO methods. Memory and runtime cost are multiples of ZO-SGD.

| Model/Task | Roberts-Large | | OPT-1.3B | | OPT-13B | | Average | Memory | Runtime |
|------------|------|--------|------|--------|------|--------|---------|--------|---------|
|            | FT   | prefix | FT   | prefix | FT   | prefix |         |        |         |
| ZO-SGD     | 89.4 | 90.0   | **90.8** | 91.4 | **90.0** | 79.0 | 88.4 | 1.0x | 1.0x |
| ZO-SGD-MMT | 89.6 | 89.1   | 85.2 | 91.2   | 87.0 | 85.0   | 87.8 | 1.56x | 1.0x |
| ZO-SGD-Cons| 89.6 | 89.1   | 88.3 | 88.1   | 82.0 | 84.0   | 86.8 | 1.0x | 2.49x |
| ZO-SGD-Sign| 52.5 | 53.6   | 87.2 | 89.5   | 80.0 | 78.0   | 73.4 | 1.0x | 1.0x |
| ZO-Adam    | 89.8 | 90.2   | 84.4 | 91.4   | 82.0 | 79.0   | 86.1 | 2.47x | 1.04x |
| HiZOO      | 93.2 | 92.7   | 90.7 | 91.4   | 88.0 | **87.0** | **90.5** | 2.04x | 1.37x |
| HiZOO-L    | 92.5 | 92.7   | 90.7 | 91.4   | 88.0 | **87.0** | 90.4 | 1.12x | 1.39x |
| FZOO       | **93.3** | **92.9** | 90.7 | **91.7** | 87.0 | **87.0** | 90.4 | 1.0x | 0.56x |

**Integration with Hybrid FO-ZO Methods.** We further demonstrate the orthogonality of FZOO by integrating it into the hybrid First-Order/Zeroth-Order framework Addax (Li et al.). By simply replacing the internal MeZO component in Addax with FZOO, we evaluated the performance on OPT-2.7B across two distinct hyperparameter settings (Setting A favors FO; Setting B favors ZO). As shown in Table 8, FZOO consistently serves as a superior "drop-in" replacement: it improves Addax's average accuracy from 72.71 to **74.11** in Setting A, and from 68.37 to **71.16** in Setting B, confirming that FZOO can effectively enhance existing hybrid strategies.

Table 8: Comparison with Hybrid FO-ZO Method (Addax) on OPT-2.7B across 11 tasks. Setting A and B denote different hyperparameters for balancing FO/ZO usage. *Average in Setting B excludes ReCoRD due to sequence length constraints.

| Method | Setting | SST-2 | RTE | CB | BoolQ | WSC | WIC | MultiRC | COPA | ReCoRD | SQuAD | DROP | Average |
|--------|---------|-------|-----|-----|-------|-----|-----|---------|------|--------|-------|------|---------|
| Addax (MeZO) | Set A | **94.2** | **81.6** | 92.9 | 75.3 | 52.9 | 64.4 | 69.4 | 81.0 | 75.2 | 84.4 | 28.6 | 72.7 |
| Addax (FZOO) | Set A | 93.1 | 78.3 | **98.2** | **76.4** | **56.7** | **66.8** | **74.1** | 81.0 | 75.8 | 84.8 | 30.0 | **74.1** |
| Addax (MeZO) | Set B | 92.2 | **79.8** | 92.9 | 63.1 | **61.5** | **61.6** | 44.5 | 82.0 | - | 81.5 | 24.6 | 68.4* |
| Addax (FZOO) | Set B | **93.6** | 76.5 | **96.4** | **75.7** | 59.6 | 57.4 | **61.4** | 84.0 | - | 82.3 | 24.6 | **71.2*** |

## 4.6 ORTHOGONALITY AND HYPERPARAMETER ANALYSIS

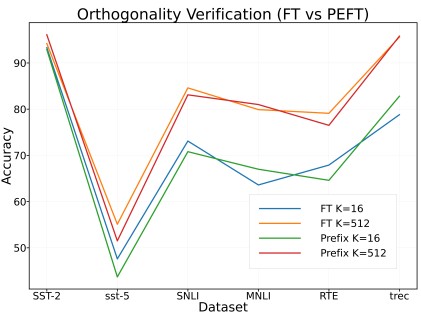
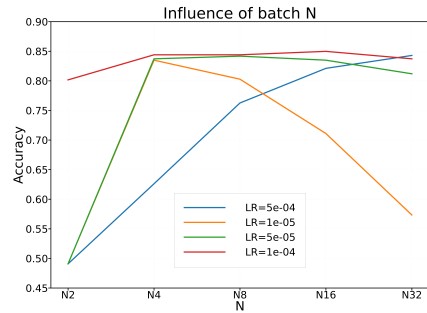

Figure 5: FZOO (prefix) vs PEFT (prefix) on Roberta-large. More results can be found in Appendix D.1.

Figure 6: Accuracy Comparison under Different Perturbation Batch Sizes N for FT. The relevant results are in Appendix G.1.

**Orthogonality to Fine-Tuning Strategies**   Our method is an optimization algorithm that focuses on how to update parameters, and is orthogonal to fine-tuning strategies that decide which parameters to update. We validate its compatibility by FT or Prefix on the roberta-large model.

**Influence of the Number of Perturbations $N$ per Step**   To further explore the trade-off between computational efficiency and convergence performance of the optimizer, we conducted a systematic evaluation on OPT-125M using the SST-2 dataset. As shown in Figure 6, increasing the number of perturbations $N$ per step can improve accuracy under certain hyperparameter settings, but also raises per-step computational cost.

## 5 CONCLUSION

We introduce FZOO, a zeroth-order optimizer that delivers order-of-magnitude Adam-like convergence speed for fine-tuning LLMs with inference-scale memory. By using batched one-sided estimates and accelerated per-batch computations, FZOO reduces forward passes and converges in far fewer steps than MeZO. It also integrates seamlessly with PEFT, offering superior performance and practical fine-tuning efficiency.

**Reproducibility Statement.**   We provide anonymous code and configuration files, together with the license and datasets, at https://anonymous.4open.science/r/FZOO-5927. Additional theoretical details appear in Appendix B. Detailed experimental settings and further results are organized in Appendices D,E,F,G, corresponding to the main experiments, efficiency analysis, and ablations, respectively.

ACKNOWLEDGMENTS

This work was supported in part by National Key Research and Development Project of China under Grant 2022YFA1004002, National Natural Science Foundation of China under Grant 72471185, and Science and Technology Project of State Grid Shaanxi Electric Power Company Limited under Grant 5226PX250001.

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

# A  RELATED WORKS

## A.1  GRADIENT-BASED OPTIMIZATION METHODS

Large language models (LLMs) have demonstrated remarkable performance across a wide range of applications (Zhao et al., 2023; Ai et al., 2025b; Zhao et al., 2025a). Standard optimizers such as SGD remain the mainstream choice for fine-tuning LLMs, but their reliance on backpropagation leads to significant memory overhead. Improvements to SGD generally fall into two categories: (1) adaptive learning rate schemes, such as AdaGrad (Duchi et al., 2011) and Adam (Kingma & Ba, 2015), and (2) accelerated schemes, such as Polyak heavy-ball (Polyak, 1964). Both approaches make use of the accumulated past gradient information to achieve faster convergence. ND-Adam (Zhang, 2018), as an improved version of Adam, significantly improves the generalization performance of the model by more precisely controlling the update direction and step size. NIGT (Cutkosky & Mehta, 2020) combines the momentum mechanism with normalized-SGD (Bernstein et al., 2018), achieving excellent empirical performance in tasks such as ResNet-50 and BERT pre-training. This method does not rely on any second-order statistics and has significantly improved memory efficiency compared to popular optimizers such as Adam. Although these methods effectively reduce some computational and storage overheads while improving adaptability and generalization capabilities, they are still essentially first-order optimization frameworks and still rely on backpropagation and gradient information.

## A.2  GRADIENT-FREE OPTIMIZATION METHODS

Gradient-free optimization estimates gradients by leveraging loss differences, thereby eliminating the need for backpropagation and intermediate activation storage (Ren et al., 2025; Yao et al., 2025a; Ren et al., 2024). With provable convergence guarantees, these methods have attracted considerable attention and have been successfully applied to a wide range of machine learning tasks, including adversarial attacks, transfer learning, and reinforcement learning, among others (Tsai et al., 2020; Shu et al., 2023; Da et al., 2025; Jing et al., 2024; Chen et al., 2023; Gu et al., 2021; Hogan & Kailkhura, 2018; Yao et al.; Zhao et al., 2025b; Ai et al., 2025a)

To reduce memory consumption during large language model fine-tuning, Malladi et al. (Malladi et al., 2023) introduced MeZO, which applies the classical ZO stochastic gradient descent algorithm Ghadimi & Lan (2013) to fine-tune pretrained LLMs without relying on backpropagation. Building on this, HiZOO Zhao et al. (2025c) incorporates diagonal Hessian information to accelerate convergence, albeit at the cost of doubled memory usage and additional hyperparameters. More recently, Sparse-MeZO Liu et al. (2024) proposes a memory-efficient ZO fine-tuning strategy that maintains memory usage comparable to standard inference. To address the issue that existing zeroth-order methods struggle to capture the low-rank gradient structures commonly found in large language model (LLM) fine-tuning—resulting in performance degradation (Chen et al., 2024; Yao et al., 2025b) proposes a low-rank ZO gradient estimator and introduces a novel algorithm, LOZO, which can effectively model such structures. Compared with the vanilla ZO-SGD method, both LOZO and its variant LOZO-M achieve improved performance. (He et al., 2024) propose a Riemannian accelerated zeroth-order algorithm with enhanced robustness, addressing the commonly observed issues of slow and unstable convergence in practical zeroth-order optimization. (Hu et al., 2024) propose a novel algorithm, namely localized zeroth-order prompt optimization (ZOPO), which incorporates a Neural Tangent Kernelbased derived Gaussian process into standard zeroth-order optimization for an efficient search of well-performing local optima in prompt optimization. However, none of the evaluated methods are yet capable of matching Adam-level speed under strict inference-level memory constraints.

## A.3  ZEROTH-ORDER OPTIMIZATION WITH STRUCTURED PERTURBATIONS

While standard ZO methods typically rely on Gaussian or spherical smoothing, several works have explored structured or directional perturbations to improve query efficiency. (Belouze, 2022) investigated the use of Rademacher random vectors for gradient estimation without backpropagation. However, their approach lacks the variance-based normalization mechanism employed in our method; without this normalization, Rademacher perturbations yield performance comparable to standard Gaussian noise used in MeZO (?), falling short of the convergence speed achieved by FZOO.

Other approaches introduce more complex structures. (Rando et al., 2024) proposed updating parameters using structured directions derived from orthonormal matrices. Although theoretically sound, this method requires constructing and storing large orthonormal matrices ($d \times N$), incurring significant memory overhead that contradicts the primary goal of memory-efficient LLM fine-tuning. Similarly, (Ma & Huang, 2025) introduced Directionally Aligned Perturbations (DAP) to minimize variance. However, DAP is specifically derived for two-point estimators and introduces additional computational overhead for vector projections ($O(d)$ operations), which prevents it from leveraging the efficient bit-level sign flips possible with Rademacher vectors in batched settings. Furthermore, (Shao & Albayrak, 2022) applied Gaussian perturbations within a zeroth-order mirror descent framework. This approach targets non-convex *composite* optimization problems, whereas LLM fine-tuning is typically formulated as a smooth non-convex problem, making the additional complexity of mirror descent unnecessary compared to our streamlined normalized update.

# B  DETAILED CONVERGENCE ANALYSIS

## B.1  PROPERTIES OF RADEMACHER RANDOM VECTOR

**Lemma B.1.** *Letting $x \in \mathbb{R}^d$ be a constant vector and $\{u_i\}_{i=1}^N$ be $d$-dimensional Rademacher random vectors, then it holds that*

$$\mathbb{E}\left[\left\|\sum_{i=1}^N u_i u_i^\top x\right\|^2\right] = N(N + d - 1)\|x\|^2. \tag{8}$$

*Proof.* It holds that

$$
\begin{aligned}
\mathbb{E}\left[\left\|\sum_{i=1}^N u_i u_i^\top x\right\|^2\right] =& \mathbb{E}\left[\sum_{i=1}^N \sum_{j=1}^N x^\top u_i u_i^\top u_j u_j^\top x\right] \\
=& \mathbb{E}\left[\sum_{j=1}^N x^\top u_j u_j^\top x \cdot \|u_j\|^2\right] + \sum_{i \neq j}\langle \mathbb{E}\left[u_i u_i^\top x\right], \mathbb{E}\left[u_j u_j^\top x\right]\rangle \\
=& \mathbb{E}\left[\sum_{j=1}^N x^\top u_j u_j^\top x \cdot \|u_j\|^2\right] + \sum_{i \neq j}\langle \mathbb{E}\left[u_i u_i^\top x\right], \mathbb{E}\left[u_j u_j^\top x\right]\rangle \\
=& Nd\|x\|^2 + (N^2 - N)\|x\|^2 \\
=& N(N + d - 1)\|x\|^2
\end{aligned}
$$

$\square$

**Lemma B.2.** *Letting $u_i$'s be Rademacher random vector, then it holds that*

$$\sum_{i=1}^N \mathbb{E}\left[\left\langle \nabla L(\theta, \mathcal{B}), u_i - \frac{1}{N}\sum_{j=1}^N u_j\right\rangle^2\right] = (N - 1)\|\nabla L(\theta, \mathcal{B})\|^2. \tag{9}$$

*Proof.* It holds that

$$\sum_{i=1}^N \left\langle \nabla L(\theta, \mathcal{B}), u_i - \frac{1}{N}\sum_{j=1}^N u_j\right\rangle^2 \tag{10}$$

$$= \sum_{i=1}^N \langle \nabla L(\theta, \mathcal{B}), u_i\rangle^2 + N\left\langle \nabla L(\theta, \mathcal{B}), \frac{1}{N}\sum_{j=1}^N u_j\right\rangle^2 \tag{11}$$

$$- 2\sum_{i=1}^N \langle \nabla L(\theta, \mathcal{B}), u_i\rangle \cdot \left\langle \nabla L(\theta, \mathcal{B}), \frac{1}{N}\sum_{j=1}^N u_j\right\rangle \tag{12}$$

$$= \sum_{i=1}^N \langle \nabla L(\theta, \mathcal{B}), u_i\rangle^2 - \frac{1}{N}\left\langle \nabla L(\theta, \mathcal{B}), \sum_{j=1}^N u_j\right\rangle^2 \tag{13}$$

Furthermore,

$$
\mathbb{E}\left[\left\langle \nabla L(\theta,\mathcal{B}), \sum_{j=1}^{N} u_j \right\rangle^2\right] = \sum_{i=1}^{N}\sum_{j=1}^{N} \langle \nabla L(\theta,\mathcal{B}), u_i \rangle \langle \nabla L(\theta,\mathcal{B}), u_j \rangle
$$

$$
= \sum_{i=1}^{N} \mathbb{E}\left[\langle \nabla L(\theta,\mathcal{B}), u_i \rangle^2\right] + \sum_{i\neq j} \mathbb{E}\left[\langle \nabla L(\theta,\mathcal{B}), u_i \rangle \langle \nabla L(\theta,\mathcal{B}), u_j \rangle\right]
$$

$$
= \sum_{i=1}^{N} \mathbb{E}\left[\langle \nabla L(\theta,\mathcal{B}), u_i \rangle^2\right]
$$

$$
= dN \left\| \nabla L(\theta,\mathcal{B}) \right\|^2.
$$

Thus,

$$
\sum_{j=1}^{N} \mathbb{E}\left[\left\langle \nabla L(\theta,\mathcal{B}), u_i - \frac{1}{N}\sum_{j=1}^{N} u_j \right\rangle^2\right] = \left(1 - \frac{1}{N}\right)\sum_{i=1}^{N} \mathbb{E}\left[\langle \nabla L(\theta,\mathcal{B}), u_i \rangle^2\right] = (N-1)\left\| \nabla L(\theta,\mathcal{B}) \right\|^2.
$$

$\square$

## B.2 PROPERTIES OF $g_t$ AND $\sigma_t$

By the Taylor's expansion, we have

$$
L(\theta + \epsilon u_i, \mathcal{B}) = L(\theta) + \epsilon \langle \nabla L(\theta,\mathcal{B}), u_i \rangle + \alpha(\theta, \epsilon u_i, \mathcal{B}) \tag{14}
$$

where $\alpha(\theta, \epsilon u_i, \mathcal{B}) \triangleq L(\theta + \epsilon u_i, \mathcal{B}) - (L(\theta) + \epsilon \langle \nabla L(\theta,\mathcal{B}), u_i \rangle)$. By the $\mathcal{L}$-smoothness, we can obtain that

$$
|\alpha(\theta, \epsilon u_i, \mathcal{B})| \leq \frac{\mathcal{L}\epsilon^2}{2}\left\| u_i \right\|^2 = \frac{d\mathcal{L}\epsilon^2}{2}, \tag{15}
$$

where the last equality is because $u_i \in \{-1, +1\}$.

**Lemma B.3.** *Letting the estimated variance $\sigma_t$ defined in equation 3, then, it holds that*

$$
\mathbb{E}\left[\sigma^2\right] = \epsilon^2 \cdot \left\| \nabla L(\theta,\mathcal{B}) \right\|^2 + \zeta, \tag{16}
$$

*with*

$$
\zeta = \frac{1}{N-1}\sum_{i=1}^{N}\left(\alpha(\theta, \epsilon u_i, \mathcal{B}) - \frac{1}{N}\sum_{j=1}^{N}\alpha(\theta, \epsilon u_j, \mathcal{B})\right)^2
$$

$$
+ \epsilon \frac{1}{N-1}\sum_{i=1}^{N}\left\langle \nabla L(\theta,\mathcal{B}), u_i - \frac{1}{N}\sum_{j=1}^{N} u_j \right\rangle \cdot \left(\alpha(\theta, \epsilon u_i, \mathcal{B}) - \frac{1}{N}\sum_{j=1}^{N}\alpha(\theta, \epsilon u_j, \mathcal{B})\right),
$$

*and*

$$
|\zeta| \leq \frac{Nd^2\mathcal{L}^2\epsilon^4}{2} + 2dN\mathcal{L}\epsilon^3 \cdot \left\| \nabla L(\theta,\mathcal{B}) \right\|.
$$

*Proof.* First, we have

$$
L(\theta + \epsilon u_i, \mathcal{B}) - \frac{1}{N}\sum_{j=1}^{N} L(\theta + \epsilon u_j, \mathcal{B})
$$

$$
\overset{equation\ 14}{=} L(\theta) + \epsilon \langle \nabla L(\theta,\mathcal{B}), u_i \rangle + \alpha(\theta, \epsilon u_i, \mathcal{B}) - \frac{1}{N}\sum_{j=1}^{N}\left(L(\theta) + \epsilon \langle \nabla L(\theta,\mathcal{B}), u_j \rangle + \alpha(\theta, \epsilon u_j, \mathcal{B})\right)
$$

$$
= \epsilon \left\langle \nabla L(\theta,\mathcal{B}), u_i - \frac{1}{N}\sum_{j=1}^{N} u_j \right\rangle + \alpha(\theta, \epsilon u_i, \mathcal{B}) - \frac{1}{N}\sum_{j=1}^{N}\alpha(\theta, \epsilon u_j, \mathcal{B}).
$$

$$\sum_{i=1}^{N} \left( L(\theta + \epsilon u_i, \mathcal{B}) - \frac{1}{N} \sum_{j=1}^{N} L(\theta + \epsilon u_j, \mathcal{B}) \right)^2$$

$$= \sum_{i=1}^{N} \left( \epsilon \left\langle \nabla L(\theta, \mathcal{B}), u_i - \frac{1}{N} \sum_{j=1}^{N} u_j \right\rangle + \beta_i \right)^2$$

$$= \sum_{i=1}^{N} \left( \epsilon^2 \left\langle \nabla L(\theta, \mathcal{B}), u_i - \frac{1}{N} \sum_{j=1}^{N} u_j \right\rangle^2 + \beta_i^2 + \epsilon \left\langle \nabla L(\theta, \mathcal{B}), u_i - \frac{1}{N} \sum_{j=1}^{N} u_j \right\rangle \beta_i \right),$$

where we use $\beta_i \triangleq \alpha(\theta, \epsilon u_i, \mathcal{B}) - \frac{1}{N} \sum_{j=1}^{N} \alpha(\theta, \epsilon u_j, \mathcal{B})$.

Furthermore, it holds that

$$\sum_{i=1}^{N} \beta_i^2 = \left( 1 - \frac{1}{N} \right) \sum_{i=1}^{N} \alpha^2(\theta, \epsilon u_i, \mathcal{B}) \leq \frac{N d^2 \mathcal{L}^2 \epsilon^4}{4},$$

and

$$\sum_{i=1}^{N} \left| \left\langle \nabla L(\theta, \mathcal{B}), u_i - \frac{1}{N} \sum_{j=1}^{N} u_j \right\rangle \beta_i \right| \leq N \left\| \left\langle \nabla L(\theta, \mathcal{B}), u_i - \frac{1}{N} \sum_{j=1}^{N} u_j \right\rangle \right\| \|\beta_i\| \overset{equation\ 15}{\leq} d N \mathcal{L} \epsilon^2 \|\nabla L(\theta, \mathcal{B})\|.$$

Therefore, we can obtain that

$$\mathbb{E}\left[\sigma^2\right] = \frac{1}{N-1} \sum_{i=1}^{N} \left( \epsilon^2 \left\langle \nabla L(\theta, \mathcal{B}), u_i - \frac{1}{N} \sum_{j=1}^{N} u_j \right\rangle^2 + \beta_i^2 + \epsilon \left\langle \nabla L(\theta, \mathcal{B}), u_i - \frac{1}{N} \sum_{j=1}^{N} u_j \right\rangle \beta_i \right)$$

$$\overset{equation\ 9}{=} \epsilon^2 \cdot \|\nabla L(\theta, \mathcal{B})\|^2 + \zeta$$

Finally, we have

$$|\zeta| \leq \frac{1}{N-1} \sum_{i=1}^{N} \beta_i^2 + \epsilon \frac{1}{N-1} \sum_{i=1}^{N} \left| \left\langle \nabla L(\theta, \mathcal{B}), u_i - \frac{1}{N} \sum_{j=1}^{N} u_j \right\rangle \beta_i \right|$$

$$\leq \frac{N d^2 \mathcal{L}^2 \epsilon^4}{2} + 2 d N \mathcal{L} \epsilon^3 \cdot \|\nabla L(\theta, \mathcal{B})\|.$$

$\square$

**Lemma B.4.** *Letting the stochastic gradient estimation $g_t$ defined in equation 2, then it satisfies the following properties*

$$g_t = \frac{1}{N} \sum_{i=1}^{N} u_i u_i^\top \nabla L(\theta_t, \mathcal{B}_t) + \tau_t \tag{17}$$

*with*

$$\tau_t = \frac{1}{N} \sum_{i=1}^{N} \epsilon^{-1} \alpha(\theta, \epsilon u_i, \mathcal{B}) \cdot u_i \text{ and } \mathbb{E}\left[\tau_t^2\right] \leq \frac{d^3 \mathcal{L}^2 \epsilon^2}{4N}. \tag{18}$$

*Proof.* By the definition of $g_t$ in equation 2, we can obtain that

$$g_t \overset{equation\ 2}{=} \frac{1}{N} \sum_{i=1}^{N} \frac{L(\theta + \epsilon u_i, \mathcal{B}) - L(\theta, \mathcal{B})}{\epsilon} u_i$$

$$\overset{equation\ 14}{=} \frac{1}{N} \sum_{i=1}^{N} u_i u_i^\top \nabla L(\theta, \mathcal{B}) + \frac{1}{N} \sum_{i=1}^{N} \epsilon^{-1} \alpha(\theta, \epsilon u_i, \mathcal{B}) \cdot u_i$$

$$= \frac{1}{N} \sum_{i=1}^{N} u_i u_i^\top \nabla L(\theta, \mathcal{B}) + \tau_t.$$

Furthermore,

$$
\begin{aligned}
\mathbb{E}\left[\tau^2\right] =& \frac{1}{N^2} \sum_{i=1}^{N} \mathbb{E}\left[\left(\epsilon^{-1}\alpha(\theta,\epsilon u_i,\mathcal{B})\right)^2 \cdot \|u_i\|^2\right] \\
=& \frac{d\epsilon^{-2}}{N} \mathbb{E}\left[\alpha^2(\theta,\epsilon u_j,\mathcal{B})\right] \overset{equation\ 15}{\leq} \frac{d^3 \mathcal{L}^2 \epsilon^2}{4N}.
\end{aligned}
$$

$\square$

**Lemma B.5.** *Letting the stochastic gradient estimation $g_t$ defined in equation 2, then its norm satisfies that*

$$
\mathbb{E}\left[\|g_t\|^2\right] = \frac{N+d-1}{N}\|\nabla L(\theta_t,\mathcal{B}_t)\|^2 + \gamma_t, \tag{19}
$$

*with*

$$
\begin{aligned}
\gamma_t =& \frac{d\epsilon^{-2}}{N}\mathbb{E}\left[\alpha^2(\theta,\epsilon u_j,\mathcal{B})\right] + \frac{d\epsilon^{-1}}{N}\mathbb{E}\left[u_i^\top \nabla L(\theta,\mathcal{B})\alpha(\theta,\epsilon u_i,\mathcal{B})\right] \\
& + \frac{(N-1)\epsilon^{-1}}{N}\left\langle \nabla L(\theta,\mathcal{B}), \mathbb{E}\left[\alpha(\theta,\epsilon u_j,\mathcal{B})\cdot u_j\right]\right\rangle.
\end{aligned}
$$

*Furthermore, it hols that*

$$
|\gamma_t| \leq \frac{d^3 \mathcal{L}^2 \epsilon^2}{4N} + d^2 \mathcal{L}\|\nabla L(\theta_t,\mathcal{B}_t)\| \cdot \epsilon.
$$

*Proof.* First, we have

$$
\begin{aligned}
\|g_t\|^2 =& \frac{1}{N^2}\left\|\sum_{i=1}^{N} u_i u_i^\top \nabla L(\theta,\mathcal{B})\right\|^2 + \frac{1}{N^2}\sum_{i=1}^{N}\left(\epsilon^{-1}\alpha(\theta,\epsilon u_i,\mathcal{B})\right)^2 \cdot \|u_i\|^2 \\
& + \frac{1}{N^2}\sum_{i=1}^{N}\sum_{j=1}^{N}\left\langle u_i u_i^\top \nabla L(\theta,\mathcal{B}), \epsilon^{-1}\alpha(\theta,\epsilon u_i,\mathcal{B})\cdot u_i\right\rangle \\
=& \frac{1}{N^2}\left\|\sum_{i=1}^{N} u_i u_i^\top \nabla L(\theta,\mathcal{B})\right\|^2 + \frac{d}{N^2}\sum_{i=1}^{N}\left(\epsilon^{-1}\alpha(\theta,\epsilon u_i,\mathcal{B})\right)^2 \\
& + \frac{1}{N^2}\sum_{i=1}^{N}\sum_{j=1}^{N}\left\langle u_i u_i^\top \nabla L(\theta,\mathcal{B}), \epsilon^{-1}\alpha(\theta,\epsilon u_j,\mathcal{B})\cdot u_j\right\rangle.
\end{aligned}
$$

We also have

$$
\mathbb{E}\left[\left\|\sum_{i=1}^{N} u_i u_i^\top \nabla L(\theta,\mathcal{B})\right\|^2\right] = N(N+d-1)\|\nabla L(\theta,\mathcal{B})\|^2.
$$

Furthermore, it holds that

$$
\begin{aligned}
& \mathbb{E}\left[\sum_{i=1}^{N}\sum_{j=1}^{N}\left\langle u_i u_i^\top \nabla L(\theta,\mathcal{B}), \epsilon^{-1}\alpha(\theta,\epsilon u_j,\mathcal{B})\cdot u_j\right\rangle\right] \\
=& \mathbb{E}\left[\sum_{i=1}^{N}\left\langle u_i u_i^\top \nabla L(\theta,\mathcal{B}), \alpha(\theta,\epsilon u_i,\mathcal{B})\cdot u_i\right\rangle\right] + \mathbb{E}\left[\sum_{i\neq j}^{N}\left\langle u_i u_i^\top \nabla L(\theta,\mathcal{B}), \epsilon^{-1}\alpha(\theta,\epsilon u_j,\mathcal{B})\cdot u_j\right\rangle\right] \\
=& Nd\epsilon^{-1}\mathbb{E}\left[u_i^\top \nabla L(\theta,\mathcal{B})\alpha(\theta,\epsilon u_i,\mathcal{B})\right] + N(N-1)\epsilon^{-1}\left\langle \nabla L(\theta,\mathcal{B}), \mathbb{E}\left[\alpha(\theta,\epsilon u_j,\mathcal{B})\cdot u_j\right]\right\rangle.
\end{aligned}
$$

Thus,

$$
\begin{aligned}
\mathbb{E}\left[\|g_t\|^2\right] =& \frac{N+d-1}{N}\|\nabla L(\theta,\mathcal{B})\|^2 + \frac{d\epsilon^{-2}}{N}\mathbb{E}\left[\alpha^2(\theta,\epsilon u_j,\mathcal{B})\right] \\
& + \frac{d\epsilon^{-1}}{N}\mathbb{E}\left[u_i^\top \nabla L(\theta,\mathcal{B})\alpha(\theta,\epsilon u_i,\mathcal{B})\right] + \frac{(N-1)\epsilon^{-1}}{N}\left\langle \nabla L(\theta,\mathcal{B}), \mathbb{E}\left[\alpha(\theta,\epsilon u_j,\mathcal{B})\cdot u_j\right]\right\rangle.
\end{aligned}
$$

Finally,

$$
\begin{aligned}
|\gamma_t| \leq & \frac{d\epsilon^{-2}}{N} \left| \mathbb{E}\left[\alpha^2(\theta, \epsilon u_j, \mathcal{B})\right] \right| + \frac{d\epsilon^{-1}}{N} \left| \mathbb{E}\left[u_i^\top \nabla L(\theta, \mathcal{B})\alpha(\theta, \epsilon u_i, \mathcal{B})\right] \right| \\
& + \frac{(N-1)\epsilon^{-1}}{N} \left| \langle \nabla L(\theta, \mathcal{B}), \mathbb{E}\left[\alpha(\theta, \epsilon u_j, \mathcal{B}) \cdot u_j\right] \rangle \right| \\
\overset{\text{equation } 15}{\leq} & \frac{d\epsilon^{-2}}{N} \cdot \frac{d^2\mathcal{L}^2\epsilon^4}{4} + \frac{d\epsilon^{-1}}{N} \left\| \nabla L(\theta_t, \mathcal{B}_t) \right\| \cdot \frac{d\mathcal{L}\epsilon^2}{2} + \epsilon^{-1} \left\| \nabla L(\theta_t, \mathcal{B}_t) \right\| \cdot \frac{d\mathcal{L}\epsilon^2}{2} \cdot d^{1/2} \\
= & \frac{d^3\mathcal{L}^2\epsilon^2}{4N} + \left( \frac{d^2\mathcal{L}\epsilon}{2N} + \frac{d^{3/2}\mathcal{L}\epsilon}{2} \right) \left\| \nabla L(\theta_t, \mathcal{B}_t) \right\| \\
\leq & \frac{d^3\mathcal{L}^2\epsilon^2}{4N} + d^2\mathcal{L} \left\| \nabla L(\theta_t, \mathcal{B}_t) \right\| \cdot \epsilon,
\end{aligned}
$$

which concludes the proof. $\qquad \square$

## B.3 PROOF OF PROPOSITION 3.2

*Proof.* Lemma B.3 and Lemma B.5 imply the result of Proposition 3.2. $\qquad \square$

## B.4 PROOF OF THEOREM 3.6

*Proof.* First, we have

$$
\mathbb{E}\left[\langle \nabla L(\theta_t), -\tau_t \rangle\right] \leq \frac{\left\| \nabla L(\theta_t) \right\|^2 + \mathbb{E}\left\| \tau_t \right\|^2}{2} \overset{\text{equation } 18}{\leq} \frac{\left\| \nabla L(\theta_t) \right\|^2}{2} + \frac{d^3\mathcal{L}^2\epsilon^2}{8N}, \tag{20}
$$

where the first inequality is because of the Cauchy's inequality.

It also holds that

$$
d\mathcal{L}^2\sigma_t^{-2}\eta^2\epsilon \cdot \left\| \nabla L(\theta_t, \mathcal{B}_t) \right\| \leq \frac{2d\mathcal{L}\sigma_t^{-2}\eta^2 \left\| \nabla L(\theta_t, \mathcal{B}_t) \right\|^2}{N} + \frac{Nd\mathcal{L}^3\sigma_t^{-2}\eta^2\epsilon^2}{2}. \tag{21}
$$

Thus, by the $\mathcal{L}$-smoothness of $\nabla L(\theta)$, we have

$$
\begin{aligned}
\mathbb{E}\left[L(\theta_{t+1})\right] \leq & L(\theta_t) - \eta\sigma_t^{-1}\mathbb{E}\left[\langle \nabla L(\theta_t), g_t \rangle\right] + \frac{\mathcal{L}\sigma_t^{-2}\eta^2}{2}\mathbb{E}\left[\left\| g_t \right\|^2\right] \\
\overset{\text{equation } 17}{=} & L(\theta_t) - \eta\sigma_t^{-1}\mathbb{E}\left[\left\langle \nabla L(\theta_t), \frac{1}{N}\sum_{i=1}^N u_i u_i^\top \nabla L(\theta_t, \mathcal{B}_t) + \tau_t \right\rangle\right] + \frac{\mathcal{L}\sigma_t^{-2}\eta^2}{2}\mathbb{E}\left[\left\| g_t \right\|^2\right] \\
= & L(\theta_t) - \eta\sigma_t^{-1}\mathbb{E}\left[\left\| \nabla L(\theta_t) \right\|^2\right] + \eta\sigma_t^{-1}\mathbb{E}\left[\langle \nabla L(\theta_t), -\tau_t \rangle\right] + \frac{\mathcal{L}\sigma_t^{-2}\eta^2}{2}\mathbb{E}\left[\left\| g_t \right\|^2\right] \\
\overset{\text{equation } 20\text{equation } 6}{\leq} & L(\theta_t) - \frac{\eta\sigma_t^{-1}}{2}\mathbb{E}\left[\left\| \nabla L(\theta_t) \right\|^2\right] + \frac{\eta\sigma_t^{-1}d^3\mathcal{L}^2\epsilon^2}{8N} + \frac{N+d-1}{N} \cdot \frac{\mathcal{L}\sigma_t^{-2}\eta^2}{2}\mathbb{E}\left[\left\| \nabla L(\theta_t, \mathcal{B}_t) \right\|^2\right] \\
& + \frac{d^3\mathcal{L}^3\sigma_t^{-2}\eta^2\epsilon^2}{8N} + d\mathcal{L}^2\sigma_t^{-2}\eta^2\epsilon \cdot \mathbb{E}\left[\left\| \nabla L(\theta_t, \mathcal{B}_t) \right\|^2\right] \\
\overset{\text{equation } 21}{\leq} & L(\theta_t) - \frac{\eta\sigma_t^{-1}}{2}\mathbb{E}\left[\left\| \nabla L(\theta_t) \right\|^2\right] + \frac{\eta\sigma_t^{-1}d^3\mathcal{L}^2\epsilon^2}{8N} + \frac{2d\mathcal{L}\sigma_t^{-2}\eta^2}{N} \cdot \mathbb{E}\left[\left\| \nabla L(\theta_t, \mathcal{B}_t) \right\|^2\right] \\
& + \frac{d^3\mathcal{L}^3\sigma_t^{-2}\eta^2\epsilon^2}{8N} + \frac{2d\mathcal{L}\sigma_t^{-2}\eta^2\mathbb{E}\left[\left\| \nabla L(\theta_t, \mathcal{B}_t) \right\|^2\right]}{N} + \frac{Nd\mathcal{L}^3\sigma_t^{-2}\eta^2\epsilon^2}{2} \\
\overset{\text{equation } 3}{\leq} & L(\theta_t) - \frac{\eta\sigma_t^{-1}}{2}\mathbb{E}\left[\left\| \nabla L(\theta_t) \right\|^2\right] + \frac{4d\mathcal{L}\sigma_t^{-2}\eta^2}{N}\mathbb{E}\left[\left\| \nabla L(\theta_t) \right\|^2\right] + \frac{4d\mathcal{L}\sigma_t^{-2}\eta^2\mathcal{V}^2}{N} \\
& + \frac{d^3\mathcal{L}^3\sigma_t^{-2}\eta^2\epsilon^2}{8N} + \frac{Nd\mathcal{L}^3\sigma_t^{-2}\eta^2\epsilon^2}{2} \\
\leq & L(\theta_t) - \frac{\eta\sigma_t^{-1}}{4}\mathbb{E}\left[\left\| \nabla L(\theta_t) \right\|^2\right] + \frac{4d\mathcal{L}\sigma_t^{-2}\eta^2\mathcal{V}^2}{N} + \frac{d^3\mathcal{L}^3\sigma_t^{-2}\eta^2\epsilon^2}{8N} + \frac{Nd\mathcal{L}^3\sigma_t^{-2}\eta^2\epsilon^2}{2},
\end{aligned}
$$

where the last inequality is because of step size satisfies $\eta \sigma_t^{-1} \leq \frac{N}{16d\mathcal{L}}$.

We can represent above equation as follows:

$$\frac{\eta \sigma_t^{-1}}{4} \mathbb{E}\left[\|\nabla L(\theta_t)\|^2\right] \leq \mathbb{E}\left[L(\theta_t) - L(\theta_{t+1})\right] + \frac{4d\mathcal{L}\sigma_t^{-2}\eta^2 \mathcal{V}^2}{N} + \frac{d^3\mathcal{L}^3\sigma_t^{-2}\eta^2\epsilon^2}{8N} + \frac{Nd\mathcal{L}^3\sigma_t^{-2}\eta^2\epsilon^2}{2}.$$

Telescoping above equation, we can obtain that

$$\frac{1}{T}\sum_{t=1}^{T}\frac{\eta\sigma_t^{-1}}{4}\mathbb{E}\left[\|\nabla L(\theta_t)\|^2\right] \leq \frac{L(\theta_1) - L(\theta^*)}{T} + \sum_{t=1}^{T}\left(\frac{4d\mathcal{L}\sigma_t^{-2}\eta^2\mathcal{V}^2}{TN} + \frac{d^3\mathcal{L}^3\sigma_t^{-2}\eta^2\epsilon^2}{8TN} + \frac{Nd\mathcal{L}^3\sigma_t^{-2}\eta^2\epsilon^2}{2T}\right).$$

Furthermore,

$$\frac{1}{T}\sum_{t=1}^{T}\mathbb{E}\left[\|\nabla L(\theta_t)\|^2\right] \leq \frac{1}{T}\sum_{t=1}^{T}\frac{\sigma_t^{-1}}{\sigma_*^{-1}}\mathbb{E}\left[\|\nabla L(\theta_t)\|^2\right]$$

$$\leq \frac{4\sigma_*(L(\theta_1) - L(\theta^*))}{\eta T} + 4\sigma_*\sum_{t=1}^{T}\left(\frac{4d\mathcal{L}\sigma_t^{-2}\eta\mathcal{V}^2}{TN} + \frac{d^3\mathcal{L}^3\sigma_t^{-2}\eta\epsilon^2}{8TN} + \frac{Nd\mathcal{L}^3\sigma_t^{-2}\eta\epsilon^2}{2T}\right)$$

By setting $\eta = \frac{1}{\sqrt{T}}$, we can obtain that

$$\frac{1}{T}\sum_{t=1}^{T}\mathbb{E}\left[\|\nabla L(\theta_t)\|^2\right] \leq \frac{4\sigma_*}{\sqrt{T}}\left(L(\theta_1) - L(\theta^*) + \frac{1}{T}\sum_{t=1}^{T}\left(\frac{4d\mathcal{L}\sigma_t^{-2}\mathcal{V}^2}{N} + \frac{d^3\mathcal{L}^3\sigma_t^{-2}\epsilon^2}{8N} + \frac{Nd\mathcal{L}^3\sigma_t^{-2}\epsilon^2}{2}\right)\right)$$

$$\tag{22}$$

$\square$

---

**Algorithm 2** FZOO-R

---

**Require:** parameters $\theta \in \mathbb{R}^d$, loss $L : \mathbb{R}^d \rightarrow \mathbb{R}$, step budget $T$, perturbation scale $\epsilon$, batch size $N$, learning rate schedule $\{\eta_t\}$

1: Initialize $l_{-1} \leftarrow \emptyset$             ▷ To store previous losses
2: **for** $t = 1, ..., T$ **do**
3:      $\ell, \theta, seeds \leftarrow$ BatchPerturbParameters$(\theta, \epsilon, N)$
4:      **if** $l_{-1}$ is not empty **then**
5:          $std \leftarrow$ standard deviation of concat$(\ell, l_{-1})$
6:      **else**
7:          $std \leftarrow$ standard deviation of $\ell$
8:      **end if**
9:      $projected\_grad \leftarrow (\ell - \mathcal{L}(\theta; \mathcal{B}))/(N * std)$
10:     $l_{-1} \leftarrow \ell$             ▷ Save the current loss
11:     BatchUpdateParameter$(projected\_grad, seeds, \theta, \eta_t)$
12: **end for**
13: **function** BATCHPERTURBPARAMETER$(\theta, \epsilon, N)$
14:      Sample batch $\mathcal{B} \subset \mathcal{D}$; obtain input $X$ and first layer weights $W^{(1)}$
15:      Initialize random seeds $seeds \leftarrow \{s_1, \ldots, s_N\}$
16:      Generate perturbation vectors $u \in \{\pm 1\}^{N \times d}$ using $seeds$
17:      Compute unperturbed activations $F^{(1)} \leftarrow W^{(1)}X$
18:      Compute perturbed activations: $Y_i^{(1)} \leftarrow F^{(1)} + \epsilon(u_i X)$, $i = 1, \ldots, N$
19:      Concatenate activations $Y^{(1)} \leftarrow [Y_1^{(1)}; \ldots; Y_N^{(1)}]$
20:      **for** $j = 2, 3, \ldots$ **do**
21:          $F^{(j)} \leftarrow W^{(j)}Y^{(j-1)}$        ▷ Compute unperturbed activations at layer $j$
22:          $P^{(j)} \leftarrow \epsilon(uY^{(j-1)})$          ▷ Compute perturbations at layer $j$
23:          $Y^{(j)} \leftarrow F^{(j)} + P^{(j)}$      ▷ Compute perturbed activations in parallel at layer $j$
24:      **end for**
25:      $\ell \leftarrow \mathcal{L}(Y^{(final)}; \mathcal{B})$        ▷ Compute final losses in parallel
26:      **return** $\ell, \theta, seeds$
27: **end function**
28: **function** BATCHUPDATEPARAMETER$(projected\_grad, seeds, \theta, \eta)$
29:      **for** $idx, s \in$ enumerate$(seeds)$ **do**
30:          Reset random number generator with seed $s$        ▷ For sampling $u$
31:          **for** $\theta_i \in \theta$ **do**
32:              $u \sim$ Uniform$(\{+1, -1\})$
33:              $\theta_i \leftarrow \theta_i - \eta * projected\_grad_{idx} * u$
34:          **end for**
35:      **end for**
36: **end function**

---

## C FZOO VARIANTS

### C.1 FZOO-R

Although FZOO accelerates training convergence through adaptive step size adjustment, each update requires multiple forward passes. To reduce the computational overhead per update, we propose a loss reuse variant FZOO-R. Instead of performing the full recommended number of forward passes for each update, we halve it. FZOO-R estimates the gradient direction solely based on these few forward passes, while reusing loss values from the previous update to adjust the step size. The detailed procedure is described in Algorithm 2. We visualize the loss curves of FZOO-R and FZOO in Figure 7. On most datasets, FZOO-R achieves comparable convergence to FZOO while requiring only half the update time per step.

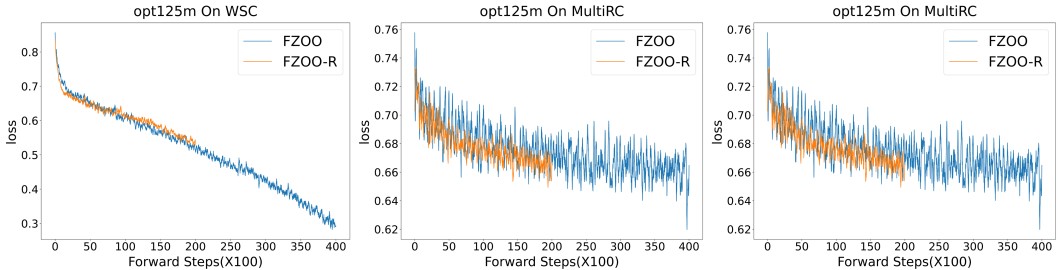

Figure 7: Loss curves on opt-125m between FZOO and FZOO-R.

## C.2  FZOO WITHOUT PARALLEL

To facilitate controlled evaluation, we release a *non-parallel* variant of FZOO, allowing for a clearer comparison of the total step-count reduction ratio against baselines such as MeZO, as illustrated in Algorithm 3. In this version, the perturbation noise is sampled from a Rademacher random vector ($\pm1$), ensuring that, apart from the removal of parallel optimization, all other components remain identical to the original FZOO. **Note.** Gaussian sampling benefits from dedicated optimized kernels, whereas PyTorch currently lacks a specialized routine for Rademacher sampling. Hence, generating Rademacher noise via `torch.randint(0, 2, size=param.data.size(), device=param.data.device, dtype=param.data.dtype) * 2 - 1` is slower than Gaussian sampling.

---

**Algorithm 3** FZOO without parallel

---

**Require:** parameters $\theta \in \mathbb{R}^d$, loss $L : \mathbb{R}^d \rightarrow \mathbb{R}$, step budget $T$, perturbation scale $\epsilon$, batch size $N$, learning rate schedule $\{\eta_t\}$

1: **for** $t = 1, ..., T$ **do**
2: $\quad \ell, \theta, seeds \leftarrow$ BatchPerturbParameters($\theta, \epsilon, N$)
3: $\quad std \leftarrow$ standard deviation of $\ell$
4: $\quad projected\_grad \leftarrow (\ell - \mathcal{L}(\theta; \mathcal{B}))/(N * std)$
5: $\quad$ BatchUpdateParameter($projected\_grad, seeds, \theta, \eta_t$)
6: **end for**
7: **function** BATCHPERTURBPARAMETER($\theta, \epsilon, N$)
8: $\quad$ Sample batch $\mathcal{B} \subset \mathcal{D}$
9: $\quad$ **for** $i = 1, ..., N$ **do**
10: $\quad\quad$ Sample random seed $s$ and append to $seeds$
11: $\quad\quad \theta \leftarrow$ PerturbParameters($\theta, \epsilon, s$)
12: $\quad\quad$ Append $\mathcal{L}(\theta; \mathcal{B})$ to $\ell$
13: $\quad\quad \theta \leftarrow$ PerturbParameters($\theta, \text{-}\epsilon, s$) $\quad\quad\quad\quad$ ▷ Reset parameters before next forward pass
14: $\quad$ **end for**
15: $\quad$ **return** $\ell, \theta, seeds$
16: **end function**
17: **function** PERTURBPARAMETER($\theta, \epsilon, s$)
18: $\quad$ Reset random number generator with seed $s$ $\quad\quad\quad\quad\quad\quad\quad\quad$ ▷ For sampling $u$
19: $\quad$ **for** $\theta_i \in \theta$ **do**
20: $\quad\quad u \sim$ Uniform($\{+1, -1\}$)
21: $\quad\quad \theta_i \leftarrow \theta_i + \epsilon u$ $\quad\quad\quad\quad\quad\quad\quad\quad\quad\quad\quad\quad$ ▷ Modify parameters in place
22: $\quad$ **end for**
23: $\quad$ **return** $\theta$
24: **end function**
25: **function** BATCHUPDATEPARAMETER($projected\_grad, seeds, \theta, \eta$)
26: $\quad$ **for** $idx, s \in$ enumerate($seeds$) **do**
27: $\quad\quad$ Reset random number generator with seed $s$ $\quad\quad\quad\quad\quad\quad$ ▷ For sampling $u$
28: $\quad\quad$ **for** $\theta_i \in \theta$ **do**
29: $\quad\quad\quad u \sim$ Uniform($\{+1, -1\}$)
30: $\quad\quad\quad \theta_i \leftarrow \theta_i - \eta * projected\_grad_{idx} * u$
31: $\quad\quad$ **end for**
32: $\quad$ **end for**
33: **end function**

---

# D EXPERIMENTS ON LLMS

## D.1 DETAILED EXPERIMENTS ON ROBERTA-LARGE

We perform FZOO experiments on the RoBERTa-large model using the hyperparameters listed in Table 9 . For learning rate scheduling and early stopping strategies, a constant learning rate is used in all FZOO experiments.

Table 9: The hyperparameter grids used for RoBERTa-large experiments. FZOO uses a constant learning rate schedule.

| Experiment | Hyperparameters | Values |
|---|---|---|
| FZOO | Batch size | 64 |
| | Learning rate | 1e−4 |
| | $\mu$ | 1e−4 |
| | Weight Decay | 0 |
| FZOO (prefix) | Batch size | 64 |
| | Learning rate | $\{1e-2, 1e-1\}$ |
| | $\mu$ | $\{1e-2, , 1e-1\}$ |
| | Weight Decay | 0 |
| | # prefix tokens | 5 |

Table 10: Experiments on RoBERTa-large (350M parameters, k=512). For MeZO we report the results we reproduced.

| Task Type | SST-2 | SST-5 | SNLI | MNLI | RTE | TREC | Average |
|---|---|---|---|---|---|---|---|
| | —— sentiment —— | | —— natural language inference —— | | | — topic — | |
| Zero-shot | 79.0 | 35.5 | 50.2 | 48.8 | 51.4 | 32.0 | 49.5 |
| LP | 91.3 | 51.7 | 80.9 | 71.5 | 73.1 | 89.4 | 76.3 |
| FT | 91.9 | 47.5 | 77.5 | 70.0 | 66.4 | 85.0 | 73.1 |
| FT (prefix) | 91.9 | 47.7 | 77.2 | 66.5 | 66.6 | 85.7 | 72.6 |
| MeZO | 93.3 | 53.2 | 83.0 | 78.3 | 78.6 | 94.3 | 80.1 |
| MeZO (prefix) | 93.3 | 53.6 | 82.9 | 75.6 | 77.2 | 88.2 | 78.4 |
| HiZOO | **95.5** | 53.2 | 82.6 | 77.7 | **80.0** | 94.6 | 80.6 |
| HiZOO (prefix) | **96.1** | **54.2** | **85.7** | 79.7 | **77.3** | 93.9 | **81.2** |
| FZOO | 94.2 | **55.1** | **84.6** | **79.9** | 79.1 | **95.6** | **81.4** |
| FZOO (prefix) | **96.1** | 51.5 | 83.1 | **81** | 76.5 | **95.8** | 80.7 |

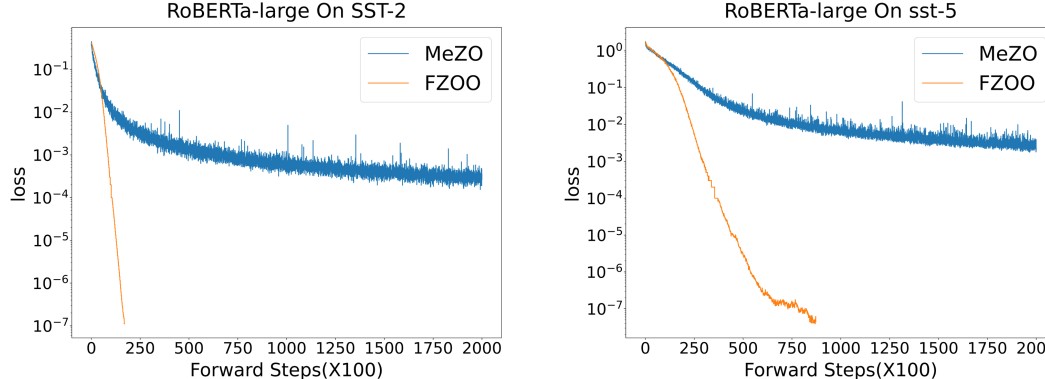

Figure 8: Loss curves on RoBERTa-large between MeZO and FZOO.

In Table 10 we show the full experiment results. Additionally, we plot more loss curves to compare with MeZO. As shown in Figure 8, we can see that FZOO can greatly accelerate the training process over MeZO, which verifies the robustness of FZOO .

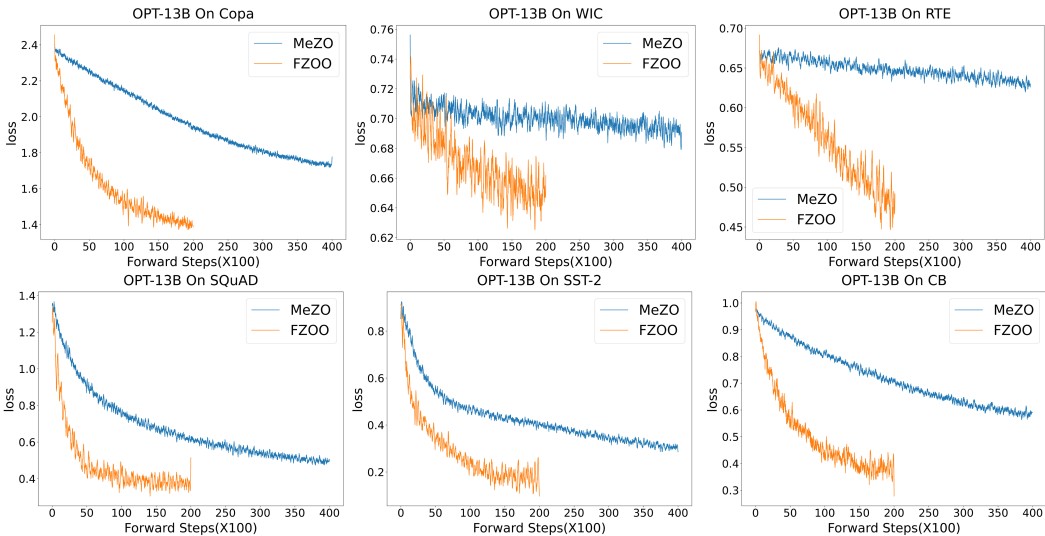

Figure 9: Loss curves on OPT.

## D.2 MORE RESULTS ON VARIOUS LLMS

We use the hyperparameters in Table 11 for FZOO experiments on OPT. Full results for OPT-2.7B and OPT-125M are in Table 12. **We also provide the relative loss curves of fine-tuning OPT-13B in Figure 9.** We also provide several loss curves of fine-tuning Phi-2(2.7B) and Llama3(8B) in Figure 10 and Figure 11.

Table 11: The hyperparameter grids used for OPT experiments. All weight decay is set to 0. FZOO uses constant learning rates.

| Experiment | Hyperparameters | Values |
|---|---|---|
| FZOO | Batch size | 16 |
| | Learning rate | $\{1e{-}5, 5e{-}5, 1e{-}4, 5e{-}4\}$ |
| | $\mu$ | $1e{-}3, 5e{-}4, 1e{-}4$ |
| FT with Adam | Batch size | 8 |
| | Learning Rates | $\{1e{-}5, 5e{-}5, 8e{-}5\}$ |

Table 12: Experiments on OPT-2.7B and OPT-125M. The best results are highlighted in bold for better comparison. We highlight the best results between FZOO and MeZO in bold to facilitate comparison.

| Model | Method | SST-2 | RTE | CB | BoolQ | WSC | WIC | MultiRC | COPA | ReCoRD | SQuAD | DROP | Average |
|---|---|---|---|---|---|---|---|---|---|---|---|---|---|
| OPT-2.7B | MeZO | 92.2 | 58.8 | 62.5 | 64.0 | 53.8 | 54.2 | 58.4 | 76.0 | 75.0 | 77.6 | 25.3 | 63.4 |
| OPT-2.7B | FZOO | **93.6** | **69.0** | 62.5 | **70.8** | **57.7** | **57.5** | **65.5** | **83.0** | **76.2** | **81.8** | **26.5** | **67.6** |
| OPT-125M | MeZO | 81.8 | 55.6 | 67.9 | 60.0 | 60.6 | 54.1 | 58.7 | 64.0 | **52.3** | 44.1 | 14.2 | 55.8 |
| OPT-125M | FZOO | **84.5** | **60.7** | 67.9 | 61.8 | 60.6 | **57.4** | 61.1 | 64.0 | 50.7 | 51.0 | 15.9 | 57.8 |

## E DETAILS ABOUT MEMORY USAGE

Here we show the detailed numbers of memory profiling results Table 13. We did not turn on any advance memory-saving options, e.g., gradient checkpointing. We set the per-device batch size as 1 to test the minimum hardware requirement to run the model with specific optimization algorithms. We use Nvidia's $nvidia-smi$ command to monitor the GPU memory usage.

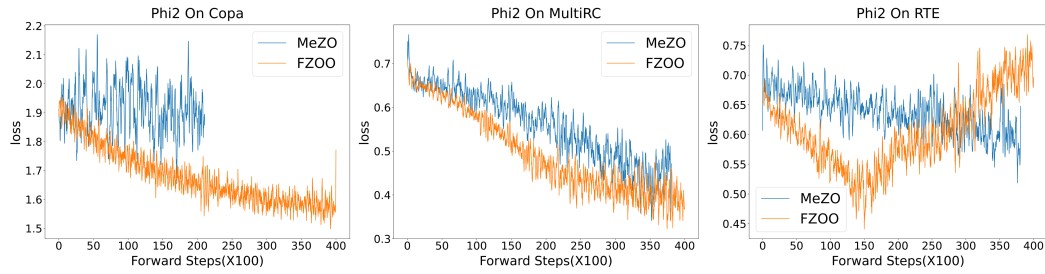

Figure 10: Loss curves on Phi-2 between MeZO and FZOO.

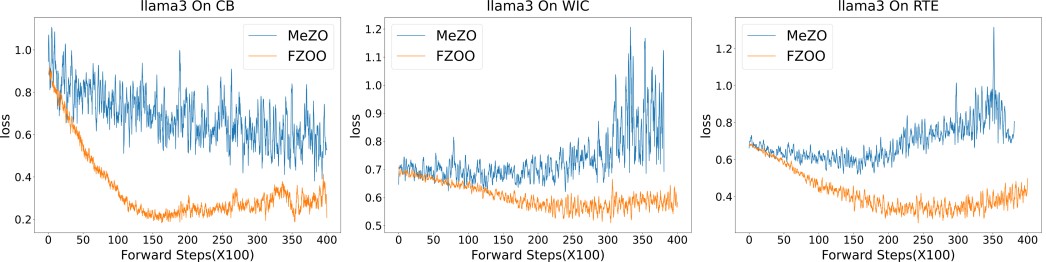

Figure 11: Loss curves on Llama3 between MeZO and FZOO.

Table 13: Memory usage on the MultiRC (average tokens=400) dataset. Results of ICL and full-parameter tuning are from MeZO(Malladi et al., 2023).

| Method | zero-shot/FZOO(FT) | HiZOO(FT) | Adam(Prefix) | ICL | Adam(FT) |
|---|---|---|---|---|---|
| 1.3B | 1xA100 (4GB) | 1xA100 (7GB) | 1xA100 (19GB) | 1xA100 (6GB) | 1xA100 (27GB) |
| 2.7B | 1xA100 (7GB) | 1xA100 (13GB) | 1xA100 (29GB) | 1xA100 (8GB) | 1xA100 (55GB) |
| 6.7B | 1xA100 (14GB) | 1xA100 (29GB) | 1xA100 (46GB) | 1xA100 (16GB) | 2xA100 (156GB) |
| 13B | 1xA100 (26GB) | 1xA100 (53GB) | 2xA100 (158GB) | 1xA100 (29GB) | 4xA100 (316GB) |
| 30B | 1xA100 (58GB) | 2xA100 (118GB) | 4xA100 (315GB) | 1xA100 (62GB) | 8xA100 (633GB) |
| 66B | 2xA100 (128GB) | 3xA100 (246GB) | 8xA100 | 2xA100 (134GB) | 16xA100 |

## F    DETAILS ABOUT WALLCLOCK TIME EFFICIENCY

As shown in Table 14, to avoid introducing additional overheads such as inter-GPU communication, results are measured on the same dataset (RoBERTa-large use SST-2 and OPT use COPA) and GPUs (80GB A100), with each result averaged over 100 steps. "BS" refers to batch size. For the relatively smaller RoBERTa-large model, we used a BS=64, while for other models, we used a BS=16.

Table 14: Wallclock time per step between Adam, MeZO and FZOO(N=8).

| Method | OPT-125M | RoBERTa-large | OPT-1.3B |
|---|---|---|---|
| Adam | 0.1982s(BS=16) | 0.3930s(BS=64) | 0.5814s(BS=16) |
| MeZO | 0.1368s(BS=16) | 0.4305s(BS=64) | 0.7218s(BS=16) |
| FZOO w/o parallel | 0.6941s(BS=16) | 1.0773s(BS=64) | 3.1925s(BS=16) |
| FZOO | 0.3835s(BS=16) | 0.6052s(BS=64) | 1.6628s(BS=16) |

## G    DETAILS ABOUT ABLATION EXPERIMENTS

### G.1    INFLUENCE OF NUMBER OF PERTURBATION DIRECTIONS USED IN EACH UPDATE

We performed an ablation study on the perturbation batch size on the SST-2 dataset using the OPT-125M model. As shown in Table 15, we observed that a batch size of 8 shows the best average performance among all tested configurations. While some other settings perform well in certain cases,

batch size 8 offers a good trade-off between stability and effectiveness, suggesting that a moderate number of perturbations per step benefits the zeroth-order optimization process.

Table 15: Ablations on OPT-125M using SST-2 dataset.

| Batch Size | (5e-5,1e-3) | (2e-4,5e-5) | (5e-4,1e-4) | (1e-5,1e-4) | (1e-4,1e-3) | (5e-5,5e-5) | (5e-5,1e-4) | Average |
|---|---|---|---|---|---|---|---|---|
| 2 | 0.8154 | 0.4908 | 0.4908 | 0.4908 | 0.7901 | 0.8165 | 0.4908 | 0.6484 |
| 4 | 0.8360 | 0.7798 | 0.6261 | **0.8349** | 0.8211 | 0.8417 | 0.8372 | 0.8026 |
| 8 | **0.8429** | 0.8521 | 0.7626 | 0.8028 | **0.8440** | 0.8326 | **0.8417** | **0.8278** |
| 16 | 0.8326 | **0.8567** | 0.8211 | 0.7110 | **0.8440** | 0.8452 | 0.8349 | 0.8244 |
| 32 | 0.7982 | 0.8498 | **0.8429** | 0.5734 | 0.8268 | 0.8154 | 0.8119 | 0.7945 |

## G.2 ABLATION ON NORMALIZATION AND PERTURBATION TYPES

To further disentangle the sources of improvement in FZOO, we conducted ablation studies to verify our design choices regarding the perturbation distribution and the normalization mechanism. As discussed in our rebuttal, the efficiency of FZOO stems from two orthogonal sources: the adaptive step size via normalization (which improves convergence rate) and the implementation of batched perturbations (which improves wall-clock time). Figure 12 visualizes these comparisons using OPT-125M on the SST-2 task.

**Gaussian vs. Rademacher Perturbations.** As shown in the left panel of Figure 12, we compare the performance of FZOO implemented with Gaussian noise versus Rademacher vectors. The loss curves indicate that the two distributions perform comparably in terms of convergence trajectory, with Rademacher achieving slightly higher final accuracy. This observation confirms that the specific choice of perturbations is not the primary factor for convergence speed. Therefore, our preference for Rademacher vectors is justified by system efficiency: as detailed in Section 3.3, the discrete nature of Rademacher vectors allows for bit-level sign flips, making batched perturbations computationally cheaper and easier to vectorize than Gaussian noise.

**Impact of Normalization.** The right panel of Figure 12 highlights the critical role of the normalization term. We compare the full FZOO algorithm against an ablation variant where the variance-based normalization is removed (i.e., updating with unnormalized estimated gradients). The results show a stark contrast: without normalization, the algorithm suffers from significantly slower convergence and struggles to minimize the loss effectively within the same iteration budget. This confirms that the normalization mechanism is the dominant component responsible for the fast convergence properties of FZOO.

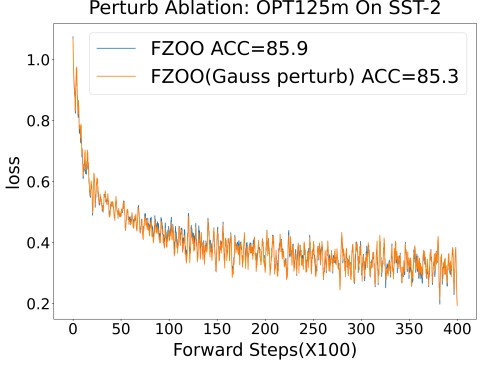 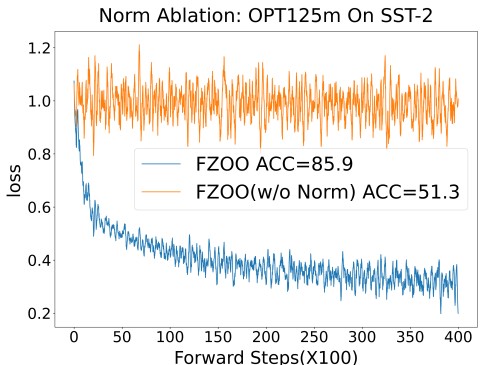

Figure 12: Ablation studies on OPT-125M (SST-2). **Left:** Effect of perturbation distribution. **Right:** Effect of variance-based normalization.

# H    DISCUSSION

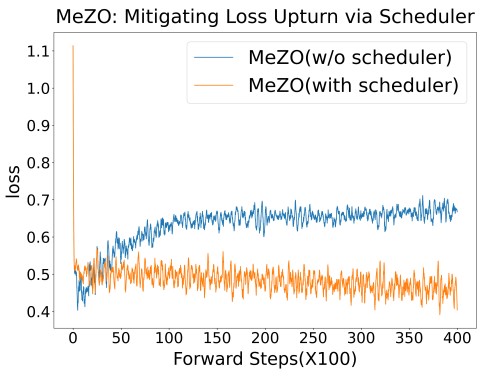 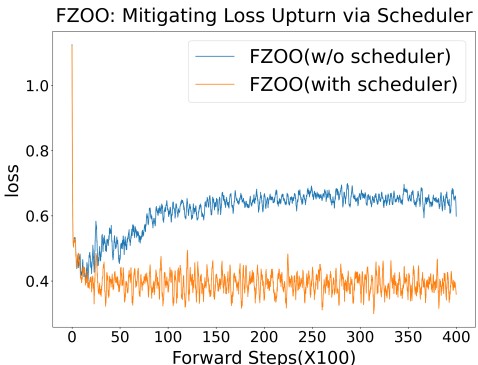

Figure 13: Loss stability analysis on OPT-125m(SST-2). Applying a cosine learning rate scheduler eliminates the late-stage loss upturn in both MeZO (Left) and FZOO (Right), confirming that the phenomenon is due to the constant step size in ZO optimization.

## H.1    TASKS WHERE FZOO LAGS FO

Across our experiments, harder generative / long-context language tasks tend to show larger gaps between ZO methods (FZOO, MeZO, HiZOO) and full first-order fine-tuning, whereas short-context classification usually shows smaller gaps. This also motivates future hybrid FO+ZO strategies (FO on a small set of sensitivity-critical parameters, ZO elsewhere).

## H.2    CONVERGENCE DISCUSSION (THEOREM 3.6) AND THE ROLE OF NORMALIZATION

**(a) Positioning Theorem 3.6 relative to existing ZO theory.**    Let $L(\theta)$ be $\mathcal{L}$-smooth. We compare the one-step descent chains of FZOO, MEZO, and HIZOO:

$$
\textbf{FZOO:}\quad \mathbb{E}[L(\theta_{t+1})] \leq L(\theta_t) - \frac{\eta\sigma_t^{-1}}{4}\left\|\nabla L(\theta_t)\right\|^2
$$
$$
+ \frac{4d\,\mathcal{L}\,\eta^2\sigma_t^{-2}\nu^2}{N} + \left(\frac{d^3\mathcal{L}^3}{8N} + \frac{Nd\mathcal{L}^3}{2}\right)\eta^2\sigma_t^{-2}\varepsilon^2, \tag{23}
$$

$$
\textbf{MeZO:}\quad \mathbb{E}[L(\theta_{t+1})] \leq L(\theta_t) - \eta\left\|\nabla L(\theta_t)\right\|^2 + \tfrac{1}{2}\eta^2\mathcal{L}\,\mathbb{E}\!\left[\left\|\nabla L(\theta_t;\mathcal{B})\right\|^2\right], \tag{24}
$$

$$
\textbf{HiZOO:}\quad \mathbb{E}[L(\theta_{t+1})] \leq L(\theta_t) - \frac{\eta}{4}\left\|\nabla L(\theta_t)\right\|_{\Sigma_t}^2 + 2\eta^2\mathcal{L}\big(\mathrm{tr}(\Sigma_t)+\beta_\omega\big)\sigma^2 + O(\mu^2). \tag{25}
$$

Ignoring the higher-order terms on the right-hand side, we want the *negative* term on the right-hand side to be as large as possible (so that the overall right-hand side is as small as possible). Hence the key comparison lies in the three terms

$$
\frac{\eta\sigma_t^{-1}}{4}\left\|\nabla L(\theta_t)\right\|^2, \qquad \eta\left\|\nabla L(\theta_t)\right\|^2, \qquad \frac{\eta}{4}\left\|\nabla L(\theta_t)\right\|_{\Sigma_t}^2.
$$

For FZOO, the effective coefficient obeys

$$
\eta\sigma_t^{-1} \ \leq\ \frac{N}{16d\mathcal{L}}.
$$

Thus a *smaller* smoothness constant $\mathcal{L}$ (a flatter region) allows a *larger* admissible value of $\eta\sigma_t^{-1}$, making the leading descent term $\frac{\eta\sigma_t^{-1}}{4}\left\|\nabla L(\theta_t)\right\|^2$ larger in magnitude and the update faster.

- **MeZO** uses the term $\eta\|\nabla L(\theta_t)\|^2$; its descent term does not automatically grow in flatter regions.
- **HiZOO** scales the gradient by the diagonal Hessian $\Sigma_t$; this curvature shaping is orthogonal to FZOO's normalization but requires storing/estimating $\Sigma_t$ and an extra forward pass.

**(b) Effect of the normalization term.** On OPT-125M in early experiments, removing the normalization (FZOO-W/O NORM) yields lower final accuracy ($\downarrow$2pp) and requires $3\times$ more forward steps to reach the same accuracy.

## H.3 $\sigma_t-\|g_t\|$ RELATION

- **Lemma B.3:**

$$\mathbb{E}\big[\|g_t\|^2\big] = \frac{N + d - 1}{N}\left\|\nabla L(\theta_t, \mathcal{B}_t)\right\|^2 + \gamma_t, \qquad \gamma_t = O(\varepsilon).$$

- **Lemma B.5:**

$$\mathbb{E}\big[\sigma_t^2\big] = \varepsilon^2 \left\|\nabla L(\theta_t, \mathcal{B}_t)\right\|^2 + \zeta_t, \qquad \zeta_t = O(\varepsilon^3).$$

Eliminating $\|\nabla L\|^2$ between Lemma B.3 and B.5 gives, to leading order,

$$\mathbb{E}\big[\sigma_t^2\big] \approx \varepsilon^2 \, \frac{N}{N + d - 1} \, \mathbb{E}\big[\|g_t\|^2\big],$$

hence

$$\sigma_t \approx \varepsilon \sqrt{\frac{N}{N + d - 1}} \, \|g_t\| \qquad \text{(up to lower-order terms)}.$$

In code we use

$$\sigma_t^2 \;=\; \|g_t\|^2 \, \varepsilon^2 \, \frac{N - 1}{N},$$

which preserves the correct scaling in $\varepsilon$, introduces a standard degrees-of-freedom correction, and works stably in practice.

## H.4 THE $\sum_{t=1}^{T} \sigma_t^{-2}$ TERM IN THEOREM 3.6

**Lemma B.3** gives

$$\mathbb{E}\big[\sigma_t^2\big] = \varepsilon^2 \, \mathbb{E}\Big[\left\|\nabla L(\theta_t, \mathcal{B}_t)\right\|^2\Big] + \zeta_t, \qquad \zeta_t > 0. \tag{26}$$

Define

$$G_{\max}^2 := \max_t \mathbb{E}\Big[\left\|\nabla L(\theta_t, \mathcal{B}_t)\right\|^2\Big], \qquad G_{\min}^2 := \min_t \mathbb{E}\Big[\left\|\nabla L(\theta_t, \mathcal{B}_t)\right\|^2\Big].$$

In stochastic optimization $G_{\min} > 0$ (Hanzely et al., 2020; Bottou et al., 2018) because a mini-batch gradient retains finite variance even at an optimum. From equation 26, for all $t$,

$$\varepsilon^2 G_{\min}^2 + \zeta_t \;\le\; \sigma_t^2 \;\le\; \varepsilon^2 G_{\max}^2 + \zeta_t. \tag{27}$$

Let $\zeta := \sup_t \zeta_t$ (finite in our setting). Then

$$\sum_{t=1}^{T} \sigma_t^{-2} \;\le\; \frac{T}{\varepsilon^2 G_{\min}^2 + \zeta}, \qquad \sigma_*^2 \;\le\; \varepsilon^2 G_{\max}^2 + \zeta. \tag{28}$$

Combining the two bounds in equation 28 yields

$$\sigma_* \sqrt{\sum_{t=1}^{T} \sigma_t^{-2}} \;\le\; \sqrt{T} \sqrt{\frac{\varepsilon^2 G_{\max}^2 + \zeta}{\varepsilon^2 G_{\min}^2 + \zeta}} \;=\; O(\sqrt{T}). \tag{29}$$

Because $\varepsilon$ is chosen small (Sec. 4.2), the constant inside the square root in equation 29 is close to 1. Inequality equation 29 shows that the product term grows at most like $\sqrt{T}$.

## H.5 ANALYSIS OF LOSS DYNAMICS AND STABILITY

We address the observation regarding the increasing loss curves in the later stages of training (e.g., Figure 3a, Figure 10c). This phenomenon is primarily attributed to the **intrinsic stochasticity** of zeroth-order optimization combined with a **constant learning rate** schedule. Unlike first-order methods where gradients vanish near local minima, ZO gradient estimates are dominated by the variance of random perturbations when the true gradient norm is small. Consequently, a constant step size can occasionally propel the model out of a sharp minimum (a "stochastic escape"), leading to a loss upturn.

We emphasize that this is a general characteristic of ZO methods rather than a specific flaw of FZOO. To validate this and demonstrate a simple mitigation, we applied a **cosine learning rate scheduler** to both MeZO and FZOO. As shown in Figure 13, the loss upturn is effectively eliminated in both methods when the learning rate decays appropriately, confirming that the instability is easily manageable without compromising the method's effectiveness.

## LLM USAGE

We used large language models only as general-purpose assist tools for minor grammar and wording suggestions. They were not involved in idea conception,experimental design or result selection. All technical content and claims were authored and verified by the authors.

