# OpenReview forum: "FZOO: Fast Zeroth-Order Optimizer for Fine‑Tuning Large Language Models towards Adam‑Scale Speed"
_ICLR.cc/2026/Conference — ICLR 2026 Poster_

### Official Review · Reviewer_jSd5 · 2025-10-29

**Soundness:** 3
**Presentation:** 3
**Contribution:** 3
**Rating:** 8
**Confidence:** 3

**Summary:**

This paper presents a new zeroth-order optimizer leveraging the idea from normalized-SGD. In comparison to the zeroth-order optimizer of MeZO,  the authors propose the following changes: (1) Generate the perturbation using Rademacher random vectors instead of a Gaussian distribution; (2) Average multiple one-sided difference estimates instead of using one two-sided difference estimate; (3) Normalize the gradient its estimated standard deviation; (4) Enable efficient implementation via batched forward pass.

The authors show that the proposed method can achieve better performance with a greatly improved the convergence rate.

**Strengths:**

1. The paper is generally well-written, clear, and, easy to follow. The proposed method is elegant, simple, but effective.

2. FZOO shows sizable improvements over MeZO in both convergence speed and model quality.

3. The authors conduct extensive experiments to support the effectiveness of the proposed method.

**Weaknesses:**

1. This paper doesn't have an extensive ablation study to test the effectiveness of each proposed change. (The ablation study only tests the effect of N, the number of perturbation directions).

**Questions:**

1. All the files under https://anonymous.4open.science/r/FZOO-5927 show the error message of "The requested file is not found.".

2. This paper doesn't include an extensive ablation study. In particular, the reviewer wonders which change contributes the most for improving the convergence speed. It seems the normalization step is the most important one. Does normalization + Gaussian perturbation work? Could the authors run further ablation studies to test the effectiveness of each proposed change?

3. It seems that averaging multiple estimates naturally enables using one-sided difference estimates. Have the authors tested the difference between averaging over N one-sided estimates v.s. averaging over N/2 two-sided estimates?

4. Could the authors elaborate a bit more on why batching recovers little speed-up for MeZO?

5. It would be useful to also compare against Addax (https://openreview.net/forum?id=QhxjQOMdDF).

---

> ### Author Response · Authors · 2025-11-22
> **Thank you for the detailed reading and valuable guidance**
>
> We greatly appreciate the reviewer’s insightful remarks. Your questions and suggestions have clarified important aspects of our method and motivated several improvements in the revised version.

---

> ### Author Response · Authors · 2025-11-22
> **W1 & Q2 — Lack of comprehensive ablation study**
>
> Thank you for the suggestion—this will make each change’s effect clearer. Our speedup comes from two **orthogonal** sources:
> (i) **adaptive step size via normalization**, which **reduces the total number of steps required for convergence**; and
> (ii) an **efficient batched forward-pass implementation**, which improves wall-clock time independent of the estimator.
>
> On your hypothesis: **normalization is the most important component.** “Normalization + Gaussian perturbation” also works. In our early OPT-125M tests, replacing Gaussian with **Rademacher** leads to similar step counts and final accuracy; we favor Rademacher because sign flips make **batched perturbations** cheaper and easier to vectorize.
>
> We will add ablations contrasting **with vs. without normalization** and **Gaussian vs. Rademacher perturbations** to quantify each contribution. **Once these ablations are finished, we will immediately update this comment with the results.**

---

> ### Author Response · Authors · 2025-11-22
> **Q1. Broken anonymous Code link**
>
> We apologize for the inconvenience. The repository was intermittently inaccessible **during roughly the last two weeks of the review period** due to a hosting-side issue. We have re-tested and—**since the start of the rebuttal phase**—the link is functioning normally. If you still encounter any access problems, please let us know and we will immediately provide a mirror and a zipped snapshot.

---

> ### Author Response · Authors · 2025-11-22
> **Q3. One-sided vs two-sided ZO averaging**
>
> Thank you for the question, we **did** run this test. A simple way to implement normalization is to repeat a **two-sided** estimator (e.g., MeZO) multiple times and use the sample **standard deviation** to set the adaptive step size. Under the **same total query budget** (i.e., averaging **N** one-sided estimates vs. **N/2** two-sided estimates), we observe **faster convergence with N one-sided averaging**. The reason is straightforward: the one-sided variant updates along **N distinct perturbation directions**, whereas the two-sided variant updates only along **N/2** directions. We will include the corresponding convergence curves in the appendix of the final version.

---

> ### Author Response · Authors · 2025-11-22
> **Q4. Why does batching offer limited speed-up for MeZO?**
>
> Thank you for the thoughtful question. In MeZO’s two-sided SPSA, a step with batch (n) requires **(2n)** forward passes (vs. **2** when (n{=}1)). With a **fixed total forward-pass budget**, batching simply trades **fewer steps** for **larger steps**, and Gaussian noise cannot take advantage of the sign-flipping property of Rademacher vector,so end-to-end wall-clock is **roughly unchanged**, and empirical gains are **small or even negative**.
>
> This conclusion is **also discussed in the MeZO paper**:
>
> > “SPSA requires only two forward passes through the model to compute the gradient estimate (for n-SPSA, each estimate requires 2n forward passes). As $\epsilon \to 0$, the SPSA estimate can be understood as a rank-1 reconstruction of the gradient. During training, (n) can be treated as a hyperparameter and follow a schedule, though in cursory experiments (Appendix A), (n=1) is the most efficient.”
>
> **The specific ablation was already conducted in the MeZO paper (Appendix A), which ultimately chose the (n=1) setting.**

---

> ### Author Response · Authors · 2025-11-22
> **Q5. Comparison with Addax**
>
> Thank you for pointing us to **Addax**; Reviewer 1YPH raised the same point. We agree it is closely related and will add a dedicated discussion in the final version.
>
> **Empirical takeaway.**
> Addax is **conceptually orthogonal** to **FZOO** and often stronger than standard first-order baselines (e.g., Adam), while FZOO remains a pure zeroth-order method. On **OPT-13B** (Adam/Addax from the Addax paper; FZOO from our runs):
>
> | Method      |   SST-2  |    RTE   |    CB    |   BoolQ  |    WSC   |    WIC   |  MultiRC |  ReCoRD  |   SQuAD  |
> | ----------- | :------: | :------: | :------: | :------: | :------: | :------: | :------: | :------: | :------: |
> | Adam        |   92.1   |   79.1   |   71.4   |   77.0   |   63.5   | **69.6** | **76.2** |   81.0   |   84.5   |
> | Addax       | **94.5** | **84.8** | **89.3** | **81.0** | **63.5** |   68.3   |   71.2   | **90.0** | **88.4** |
> | FZOO (ours) |   93.7   |   71.1   |   69.6   |   72.2   |   63.5   |   60.5   |   66.0   |   81.0   |   84.8   |
>
> Addax’s public repo currently lacks a complete environment file, so a full reproduction is ongoing; hence the Adam/Addax numbers above are taken from their paper.
>
> **Methodological relationship (orthogonality).**
>
> * **Addax:** mixes FO and ZO at the **data-batch level**. IP-SGD (FO) for short sequences, MeZO (ZO) for long—without changing the underlying FO/ZO optimizers.
> * **FZOO:** improves the **ZO component itself** (normalization + Rademacher perturbations) while keeping memory at **inference level** (no backprop/gradient storage).
>
> Thus, wherever Addax uses MeZO for the long-sequence branch, one could in principle swap in **FZOO** to form an “**IP-SGD + FZOO**” hybrid. Once we finish a faithful Addax setup, we plan to implement this variant and submit an update comment.
>
> **Efficiency considerations.**
> Addax gains speed from FO updates on short inputs but uses **more memory** than a pure ZO pipeline (due to activation/gradient storage on the FO branch). FZOO uses ZO for both short and long inputs, so training memory matches **inference-time memory**, but it cannot leverage FO gradients. We will clarify this design trade-off and position Addax as **complementary** hybrids that can be combined with FZOO in future work.

---

> > ### Comment · Reviewer_ovaX · 2025-11-25
> >
> > I thank the authors for their detailed reply, which addressed some of my concerns. The primary clarification that the paper is not focused on theoretical analysis is helpful.
> >
> > I have some follow-up questions about the algorithm's stability. In Figures 3a, 10c, and 11, we observe increasing loss during the later phase of training. It would be helpful if the author could provide some detailed discussion on this behavior.

---

> > > ### Author Response · Authors · 2025-11-26
> > > **Response to Follow-up: Clarification on Loss Dynamics and Stability**
> > >
> > > Thank you for your continued engagement and for raising this insightful question regarding the **increasing loss curves observed in the later stages of training** (Figures 3a, 10c, 11).
> > >
> > > This phenomenon is an expected behavior given the optimization dynamics of Zeroth-Order methods, though it is worth noting that **this behavior is stochastic and does not occur in every trial**. We provide a detailed explanation from three perspectives:
> > >
> > > **1. Intrinsic Stochasticity of ZO Optimization**
> > > Unlike First-Order methods (e.g., Adam) which utilize momentum and adaptive moment estimation to naturally decay the effective step size, ZO methods (like MeZO and FZOO) often utilize a constant learning rate schedule in fine-tuning tasks.
> > > As the model approaches a sharp local minimum, the true gradient norm approaches zero, making the ZO gradient estimate dominated by the variance of the random perturbation.
> > > * **Stochastic "Escape":** This loss increase is not deterministic. It occurs probabilistically when a specific random perturbation, combined with a non-decaying learning rate, accidentally propels the model parameters out of the narrow valley of the optimal solution. In many other runs (where the random seed differs), the loss may remain stable.
> > > * **Trade-off:** This is a common trade-off in ZO optimization where ensuring stability at the very end of training would require a strictly decaying scheduler, which might conversely slow down the initial convergence.
> > >
> > > **2. FZOO Specific Dynamics**
> > > Regarding FZOO specifically, while it uses the estimated standard deviation $\sigma_t$ to normalize updates ($\eta_t / \sigma_t$), this mechanism is primarily designed to accelerate convergence in the early-to-mid stages. Near convergence, $\sigma_t$ stabilizes at a small non-zero value (representing the intrinsic noise floor), meaning the effective step size does not vanish automatically. Consequently, once FZOO rapidly reaches the optimum (often faster than baselines), continuing to train with the same step size increases the probability of the stochastic "escape" described above, leading to the observed loss upturn.
> > >
> > > **3. Practical Impact and Mitigation**
> > > In practice, this late-stage behavior does not negatively impact the model's utility.
> > > * **Early Stopping:** All reported accuracy metrics in our paper are derived from the **best evaluation checkpoint**, effectively applying Early Stopping. The "rising tail" of the loss curve occurs *after* the model has already achieved peak performance.
> > > * **Convergence Speed:** The fact that this rise is observable in the figures actually reflects FZOO's superior convergence speed—it reaches the optimum potential of the current learning rate much earlier than the baseline.
> > >
> > > **4. Manuscript Revision Plan**
> > > Since the comment system does not support image uploads, we plan to include a detailed discussion of this phenomenon in **Appendix H.6** of our revised manuscript. Specifically, we will add illustrative figures demonstrating:
> > > * (1) **MeZO also exhibits this stochastic loss upturn**, confirming it is a general ZO characteristic rather than an FZOO-specific flaw.
> > > * (2) **Applying a suitable learning rate scheduler prevents the rise**, validating our explanation regarding the step size.
> > > We will upload the updated PDF containing these additions before the end of the rebuttal period, once all concurrent experiments are finalized.
> > >
> > > **Summary**
> > > The increasing loss is a byproduct of the fixed learning rate schedule interacting with the stochastic noise of ZO estimation after convergence, rather than a fundamental instability in the FZOO algorithm itself. It can be easily managed via early stopping without compromising peak performance.
> > >
> > > **Closing Remark**
> > > We deeply appreciate the time you have taken to re-examine our empirical results. Since our previous response addressed the theoretical clarification, and this response clarifies the stability of the training dynamics, we believe the major concerns regarding the paper's validity have been resolved. If you find these explanations satisfactory, **we would be grateful if you could reconsider your evaluation to reflect these clarifications.** We remain fully available should you have any further questions.

---

> > ### Author Response · Authors · 2025-11-26
> > **Follow-up to Q5. Comparison with Addax**
> >
> > As a follow-up to our previous discussion regarding the relationship with Addax, we are pleased to report that we have completed the full reproduction and comparison experiments. These results empirically confirm our hypothesis that FZOO is orthogonal to hybrid FO-ZO frameworks.
> >
> > **Experimental Configurations**
> > To complete the verification within the rebuttal timeframe, we performed experiments on **OPT-2.7B**. We evaluated the two hyperparameter settings found in the official Addax repository to ensure robustness:
> > 1.  **Setting A (README):** $K_0=2, K_1=10, L_T=317, \alpha=1\text{e-}3$. Heavily utilizes the First-Order component (IP-SGD).
> > 2.  **Setting B (Appendix):** $K_0=6, K_1=4, L_T=160, \alpha=5\text{e-}4$. Relies more heavily on the Zeroth-Order component.
> >
> > We constructed an **"Addax + FZOO"** variant by strictly substituting the MeZO optimizer with FZOO for the long-sequence processing branch, leaving all other mechanisms (e.g., sequence sorting, batching strategies) unchanged.
> >
> > **Comparison Results**
> > The table below details the Average Accuracy across 11 tasks. **Bold** highlights the superior method in each comparison.
> >
> > | Method | Hyperparams | SST-2 | RTE | CB | BoolQ | WSC | WIC | MultiRC | COPA | ReCoRD | SQuAD | DROP | **AVG** |
> > | :--- | :--- | :--- | :--- | :--- | :--- | :--- | :--- | :--- | :--- | :--- | :--- | :--- | :--- |
> > | **MeZO (Pure ZO FT)** | EPS=1e-3 | 92.2 | 58.8 | **62.5** | 64.0 | 53.8 | 54.2 | 58.4 | 76.0 | **75.0** | 77.6 | 25.3 | 63.44 |
> > | **FZOO (Pure ZO FT)** | EPS=1e-4 | **93.6** | **69.0** | **62.5** | **70.8** | **57.7** | **57.5** | **65.5** | **83.0** | 74.8 | **81.8** | **26.5** | **67.52** |
> > | | | | | | | | | | | | | | |
> > | **Addax (MeZO)** | Setting A | **94.15** | **81.59** | 92.86 | 75.3 | 52.88 | 64.42 | 69.4 | 81.0 | 75.2 | 84.42 | 28.56 | 72.71 |
> > | **Addax (FZOO)** | Setting A | 93.12 | 78.34 | **98.21** | **76.4** | **56.73** | **66.77** | **74.1** | 81.0 | **75.8** | **84.78** | **29.99** | **74.11** |
> > | | | | | | | | | | | | | | |
> > | **Addax (MeZO)** | Setting B | 92.2 | **79.78** | 92.86 | 63.1 | **61.54** | **61.6** | 44.5 | 82.0 | - | 81.52 | 24.57 | 68.37* |
> > | **Addax (FZOO)** | Setting B | **93.58** | 76.53 | **96.43** | **75.7** | 59.62 | 57.37 | **61.4** | **84.0** | - | **82.31** | **24.61** | **71.16*** |
> >
> > *\*Note: In Setting B, the stricter sequence length constraint ($L_T=160$) caused runtime errors for the ReCoRD task (due to input length exceeding the FO limit), requiring its exclusion from the average. This limitation aligns with the original Addax paper, which also omitted ReCoRD and DROP results.*
> >
> > **Conclusion**
> > Our experiments confirm that FZOO acts as a **superior "plug-and-play" module** for hybrid methods:
> > * Under **Setting A**, integrating FZOO boosts the average performance from 72.71 to **74.11**.
> > * Under **Setting B**, the gain is even more pronounced, rising from 68.37 to **71.16**.
> >
> > This validates that FZOO is complementary to data-batching strategies like Addax. By upgrading the underlying ZO optimizer from MeZO to FZOO, we can achieve strictly better performance without altering the hybrid framework's architecture.

---

### Official Review · Reviewer_ovaX · 2025-10-31

**Soundness:** 3
**Presentation:** 3
**Contribution:** 2
**Rating:** 2
**Confidence:** 2

**Summary:**

This paper studies the memory and convergence tradeoff in zeroth-order optimization. The paper proposes FZOO (Fast Zeroth-Order Optimizer), a novel optimization method designed to fine-tune large language models with inference-level memory usage while achieving Adam-like convergence speed. The method combines batched Rademacher random perturbation with adaptive learning rate to mimic normalized-SGD (and Adam) behavior. The paper shows that the proposed algorithm converges in $O(\sqrt{d}/\sqrt{T})$ by properly choosing learning rate and perturbation stepsize.

Numerical results shows that the proposed method outperforms existing MeZO method in both final accuracy and convergence speed.

**Strengths:**

1. The paper proposed a novel approach to compute the activations for the perturbed model with Rademacher perturbation. The proposed method speeds up the loss computation.
2. The paper proposed using the batched loss variance to normalize the stepsize, which enables accelerated convergence speed.
3. The paper demonstrated promising numerical results showing FZOO outperforms MeZO and HiZOO in multiple settings.

**Weaknesses:**

1. Missing discussion on the effect of normalization with estimated variance. The author should further compare the convergence results with and without normalization. The current discussion in section 3.4 is insufficient to demonstrate the usefulness of the normalization. As we know, normalized SGD actually fails to converge to a stationary solution in specific problems.

2. Missing reference to existing ZO algorithms that use structured/directional perturbation. E.g., [R1-R4]. The proof technique is standard for ZOO with either Normal or Rademacher random perturbation.

3. The paper should compare the proposed FZoo method with other structural perturbation methods or variance-reduced methods.

[R1] Belouze, G. Optimization without backpropagation, 2022. URLhttps://arxiv.org/abs/2209.06302.

[R2] Rando M, Molinari C, Villa S, Rosasco L. Stochastic zeroth order descent with structured directions. Computational Optimization and Applications. 2024 Dec;89(3):691-727.

[R3] Ma S, Huang H. Revisiting zeroth-order optimization: Minimum-variance two-point estimators and directionally aligned perturbations. In The Thirteenth International Conference on Learning Representations 2025.

[R4] Shao, W., Albayrak, S. (2023). Adaptive Zeroth-Order Optimisation of Nonconvex Composite Objectives. In: Nicosia, G., et al. Machine Learning, Optimization, and Data Science. LOD 2022

**Questions:**

Please address the above weakness, especially the comparison with other structural perturbation methods.

Also, in the equations on page 20, I think some expectations are missing.

The final steps on page 21 (from line 1904 to lines 1104) do not look correct to me. I think a V term is missing in the first term, and thus, the main theorem 3.6 is incorrect.

By having the term $\sigma_*/\sqrt{(\bar{\sigma}_t^{-2})^{1/2}} \geq 1$, it seems like the normalization is slowing down the convergence?

---

> ### Author Response · Authors · 2025-11-21
> **Thank you for the insightful feedback that improved our work**
>
> We are grateful for the reviewer’s detailed critique and thoughtful suggestions. We have carefully addressed all concerns and believe the manuscript has benefited substantially from your feedback.

---

> ### Author Response · Authors · 2025-11-21
> **W1. Missing discussion on the effect of normalization with estimated variance**
>
> Please note that our paper is **not** intended as a theoretical work proving that normalized SGD always outperforms standard SGD in general. Instead, motivated by the empirical success of various *normalized SGD–type* methods in training deep neural networks (e.g., Path-SGD[1], mean-normalized SGD[2], and stochastic normalized gradient descent with momentum[3]), we design FZOO as a *zeroth-order analogue* of normalized SGD to achieve fast convergence in LLM fine-tuning.
>
> In this view, MeZO can be regarded as a zeroth-order method **without** normalization, while FZOO adds a variance-based normalization step along the sampled direction. Therefore, our experimental comparison between FZOO and MeZO **already provides a direct comparison of convergence behavior with and without normalization** in the zeroth-order setting.
>
> We fully agree that normalized SGD is known to fail to converge to a stationary solution on certain specific pathological problems. Our goal here is more modest: we only claim and demonstrate that, on the LLM fine-tuning tasks considered in our experiments, the normalized ZO update in FZOO behaves well and improves convergence in practice. Thanks to your reminder, we will explicitly clarify this scope in the revision and will be more cautious in discussing the theoretical limitations of normalization-based methods.
>
> [1] Neyshabur, Behnam, Russ R. Salakhutdinov, and Nati Srebro. "Path-sgd: Path-normalized optimization in deep neural networks." Advances in neural information processing systems 28 (2015).
>
> [2] Wiesler, Simon, et al. "Mean-normalized stochastic gradient for large-scale deep learning." 2014 IEEE International Conference on Acoustics, Speech and Signal Processing (ICASSP). IEEE, 2014.
>
> [3] Zhao, Shen-Yi, et al. "Stochastic normalized gradient descent with momentum for large-batch training." Science China Information Sciences 67.11 (2024): 212101.

---

> ### Author Response · Authors · 2025-11-21
> **W2. Missing reference to existing ZO algorithms with structured/directional perturbations**
>
> Thank you for pointing out these important related works. We will add [R1–R4] in the revised version and explicitly discuss how our method relates to existing ZO algorithms that use structured or directional perturbations.

---

> ### Author Response · Authors · 2025-11-21
> **W3 and Q1**
>
> ## [R1]
> Work in [R1] also uses Rademacher random vectors, similar to our work.
> However, without normalization by $\sigma_t^{-1}$, the Rademacher random vectors only achieve performance similar to using Gaussian noise as in MeZO [4].
> Similar performance is also achieved when using spherical perturbations.
> Accordingly, these methods do not achieve performance comparable to FZOO on our benchmarks.
> ## [R2]
> In [R2], an orthonormal matrix $P \in \mathbb{R}^{d \times N}$ with $N$ being the batch size is used to update the parameters as
> $$
>     \theta_{t+1} = \theta_t - \eta P P^\top \nabla L(\theta, \mathcal{B}).
> $$
> However, this kind of structured perturbation is difficult to apply in LLM fine-tuning.
> Because $P \in \mathbb{R}^{d \times N}$ is required to be orthonormal, one has to store the matrix $P$, which incurs a large memory cost.
> One important purpose of zeroth-order optimization in our setting is to save memory.
> In contrast, our FZOO only needs to store random seeds and generate the Rademacher vectors when needed.
> ## [R3]
> For [R3], Their DAP estimator is derived specifically for the **two-point** zeroth-order setting, and does not directly extend to the **multi-point** estimation paradigm used in FZOO. DAP introduces extra overhead compared with simple Rademacher perturbations: it requires maintaining a gradient-direction vector ((O(d)) memory) and performing vector projections/normalizations at each step ((O(d)) operations). While this cost is negligible relative to LLM forward passes, it prevents DAP from enjoying the key advantage of Rademacher noise in batched settings, where perturbations reduce to cheap sign flips. Experimentally, DAP is only evaluated on OPT-1.3B with SST-2, where it indeed outperforms Gaussian perturbations but still underperforms FZOO. Moreover, on OPT-1.3B with SST-2, their results also show that **Rademacher perturbations already outperform Gaussian noise**.
> ## [R4]
> [R4] also uses Gaussian perturbations, but to solve nonconvex **composite** problems via zeroth-order mirror descent.
> Our LLM fine-tuning problem is not a nonconvex composite problem, but rather a nonconvex smooth problem.
> Thus zeroth-order mirror descent is not required for our LLM fine-tuning setting.
>
> [4] Malladi, Sadhika, et al. "Fine-tuning language models with just forward passes." Advances in Neural Information Processing Systems 36 (2023): 53038-53075.
>
> [R1] Belouze, G. Optimization without backpropagation, 2022. URL https://arxiv.org/abs/2209.06302.
>
> [R2] Rando M, Molinari C, Villa S, Rosasco L. Stochastic zeroth order descent with structured directions. Computational Optimization and Applications. 2024 Dec;89(3):691-727.
>
> [R3] Ma S, Huang H. Revisiting zeroth-order optimization: Minimum-variance two-point estimators and directionally aligned perturbations. In The Thirteenth International Conference on Learning Representations 2025.
>
> [R4] Shao, W., Albayrak, S. (2023). Adaptive Zeroth-Order Optimisation of Nonconvex Composite Objectives. In: Nicosia, G., et al. Machine Learning, Optimization, and Data Science. LOD 2022

---

> ### Author Response · Authors · 2025-11-21
> **Q2 & Q3. Questions about the proofs**
>
> Thank you for your kind reminder. We have added the missing expectation notations and fixed several typos in the derivations.
> The revised version of the proof already have been included in the updated paper with \[UPDATE\] tag.
>
> The original lines 294-310 have been modified as follows:
> **Theorem 3.6 (Convergence of FZOO).**
> Let the objective function $L(\theta)$ be $\mathcal{L}$-smooth, and Assumption~3.5 hold. The update rule is $\theta_{t+1} = \theta_t - \eta_t g_t$, where the effective step size $\eta_t = \frac{\eta}{\sigma_t}$ satisfies $\eta_t \leq \frac{N}{16d\mathcal{L}}$, and the learning rate factor $\eta$ is set as
> $ \eta = \frac{1}{\sqrt{T}}$.
> Then, after $T$ iterations, the FZOO algorithm satisfies:
>
> $$
>     \frac{1}{T}\sum_{t=1}^{T} \mathbb{E}\left[ \left|\left|{\nabla L(\theta_t)}\right|\right|^{2} \right]
>     \leq
>     \frac{4\sigma_*}{\sqrt{T}} \left( L(\theta_1) - L(\theta^*) + \frac{1}{T}\sum_{t=1}^{T}\left(\frac{4d \mathcal{L} \sigma_t^{-2} \mathcal{V}^2}{N}
> +\frac{d^3\mathcal{L}^3\sigma_t^{-2} \epsilon^2}{8N} + \frac{Nd\mathcal{L}^3\sigma_t^{-2}\epsilon^2}{2}\right) \right)
> $$
>
> where $\theta^*$ denotes the optimal solution (minimum) of $L(\theta)$.
>
> And $\sigma_* = \max_{t \in \{1, \dots, T\}} \sigma_t$
>
> This theorem shows that the FZOO algorithm converges to a stationary point whose gradient norm close to zero.
> Especially, our theorem shows that to find obtain $\frac{1}{T}\sum_{t=1}^{T} \mathbb{E}\left[ \left|\left|{\nabla L(\theta_t)}\right|\right|^{2}\right] \leq \varepsilon$ with $\varepsilon>0$, it requires $T = O(\varepsilon^{-2})$ which matches the convergence rate of stochastic gradient descent for nonconvex optimization.
>
> The original lines 1096-1112(current line 1097-1111) have been modified as follows:
> By setting $\eta = \frac{1}{\sqrt{T}}$,
> we can obtain that
> $$
>     \frac{1}{T}\sum_{t=1}^{T} \mathbb{E}\left[ \left|\left|{\nabla L(\theta_t)}\right|\right|^{2}\right]
>     \leq
>     \frac{4\sigma_*}{\sqrt{T}} \left( L(\theta_1) - L(\theta^*) + \frac{1}{T}\sum_{t=1}^{T}\left(\frac{4d \mathcal{L} \sigma_t^{-2} \mathcal{V}^2}{N}
> +\frac{d^3\mathcal{L}^3\sigma_t^{-2} \epsilon^2}{8N} + \frac{Nd\mathcal{L}^3\sigma_t^{-2}\epsilon^2}{2}\right) \right)
> $$

---

> ### Author Response · Authors · 2025-11-21
> **Q4. Is normalization slowing down the convergence?**
>
> Our work is not a theory paper.
> The theoretical analysis is only intended to show that FZOO converges, rather than to prove that FZOO must outperform MeZO in the sense of convergence rates.
> In fact, to the best of our knowledge, there is no existing work that can theoretically prove that normalized SGD always outperforms standard SGD.

---

> ### Author Response · Authors · 2025-11-26
> **Response to Follow-up: Clarification on Loss Dynamics and Stability**
>
> Thank you for your continued engagement and for raising this insightful question regarding the **increasing loss curves observed in the later stages of training** (Figures 3a, 10c, 11).
>
> This phenomenon is an expected behavior given the optimization dynamics of Zeroth-Order methods, though it is worth noting that **this behavior is stochastic and does not occur in every trial**. We provide a detailed explanation from three perspectives:
>
> **1. Intrinsic Stochasticity of ZO Optimization**
> Unlike First-Order methods (e.g., Adam) which utilize momentum and adaptive moment estimation to naturally decay the effective step size, ZO methods (like MeZO and FZOO) often utilize a constant learning rate schedule in fine-tuning tasks.
> As the model approaches a sharp local minimum, the true gradient norm approaches zero, making the ZO gradient estimate dominated by the variance of the random perturbation.
> * **Stochastic "Escape":** This loss increase is not deterministic. It occurs probabilistically when a specific random perturbation, combined with a non-decaying learning rate, accidentally propels the model parameters out of the narrow valley of the optimal solution. In many other runs (where the random seed differs), the loss may remain stable.
> * **Trade-off:** This is a common trade-off in ZO optimization where ensuring stability at the very end of training would require a strictly decaying scheduler, which might conversely slow down the initial convergence.
>
> **2. FZOO Specific Dynamics**
> Regarding FZOO specifically, while it uses the estimated standard deviation $\sigma_t$ to normalize updates ($\eta_t / \sigma_t$), this mechanism is primarily designed to accelerate convergence in the early-to-mid stages. Near convergence, $\sigma_t$ stabilizes at a small non-zero value (representing the intrinsic noise floor), meaning the effective step size does not vanish automatically. Consequently, once FZOO rapidly reaches the optimum (often faster than baselines), continuing to train with the same step size increases the probability of the stochastic "escape" described above, leading to the observed loss upturn.
>
> **3. Practical Impact and Mitigation**
> In practice, this late-stage behavior does not negatively impact the model's utility.
> * **Early Stopping:** All reported accuracy metrics in our paper are derived from the **best evaluation checkpoint**, effectively applying Early Stopping. The "rising tail" of the loss curve occurs *after* the model has already achieved peak performance.
> * **Convergence Speed:** The fact that this rise is observable in the figures actually reflects FZOO's superior convergence speed—it reaches the optimum potential of the current learning rate much earlier than the baseline.
>
> **4. Manuscript Revision Plan**
> Since the comment system does not support image uploads, we plan to include a detailed discussion of this phenomenon in **Appendix H.6** of our revised manuscript. Specifically, we will add illustrative figures demonstrating:
> * (1) **MeZO also exhibits this stochastic loss upturn**, confirming it is a general ZO characteristic rather than an FZOO-specific flaw.
> * (2) **Applying a suitable learning rate scheduler prevents the rise**, validating our explanation regarding the step size.
> We will upload the updated PDF containing these additions before the end of the rebuttal period, once all concurrent experiments are finalized.
>
> **Summary**
> The increasing loss is a byproduct of the fixed learning rate schedule interacting with the stochastic noise of ZO estimation after convergence, rather than a fundamental instability in the FZOO algorithm itself. It can be easily managed via early stopping without compromising peak performance.
>
> **Closing Remark**
> We deeply appreciate the time you have taken to re-examine our empirical results. Since our previous response addressed the theoretical clarification, and this response clarifies the stability of the training dynamics, we believe the major concerns regarding the paper's validity have been resolved. If you find these explanations satisfactory, **we would be grateful if you could reconsider your evaluation to reflect these clarifications.** We remain fully available should you have any further questions.

---

### Official Review · Reviewer_pkyA · 2025-11-01

**Soundness:** 3
**Presentation:** 3
**Contribution:** 3
**Rating:** 6
**Confidence:** 3

**Summary:**

The paper introduces FZOO, a zeroth‑order optimizer for LLM fine‑tuning that (i) uses batched one‑sided function evaluations and an adaptive step size normalized by the batch loss standard deviation, linking its update to normalized‑SGD, (ii) accelerates each step via Rademacher (±1) perturbations that enable efficient batched parallelism. Experiments show FZOO typically beats MeZO by ~+3% accuracy while using ~3× fewer forward passes across various datasets and tasks. On RoBERTa‑large reports Adam‑like convergence at inference‑level memory, with accompanying proofs of normalized‑SGD equivalence and convergence guarantees.

**Strengths:**

1. FZOO brings normalized‑SGD’s normalization into the ZO regime by scaling updates with the batch loss standard deviation and using one‑sided estimates, making ZO steps both more stable and step‑efficient.

2. The paper proves a formal link to normalized‑SGD and a convergence guarantee under standard smoothness/variance assumptions

3. Rademacher (±1) perturbations enable per‑layer sign‑flip/add operations and batched parallelism, yielding speed‑up.

4. Strong empirical gains across scales and tasks, and Inference‑level memory footprint

**Weaknesses:**

1. 3.3 argues Rademacher allows “bit‑level” sign flips so additions replace multiplies, there is no roofline or kernel profile to substantiate “addition beats multiply” benefits at scale.

2. The paper compares to prefix‑tuning, but omits head‑to‑head QLoRA/LoRA+Adam under matched memory/throughput budgets, which are widely used and could potentially alter the claimed end‑to‑end efficiency picture.

**Questions:**

1. Theorem 3.6 requires d framed as “parameter dimension”, what is the correct setting?

2. The advertised 1.92× speedup (OPT‑125M, N=8) compares against an 8‑perturbation sequential baseline, realistic strong baselines like MeZO with N=1 or two‑sided ZO with efficient batching. what causes this inflation?

---

> ### Author Response · Authors · 2025-11-21
> **Thank you for the careful review and helpful suggestions**
>
> We truly appreciate the reviewer’s careful reading and insightful feedback. Your comments have been extremely valuable in strengthening the presentation of our work.

---

> ### Author Response · Authors · 2025-11-21
> **W1. Lack of evidence for “addition beats multiply” claim in Section 3.3**
>
> Thank you for the thoughtful question. Replacing Gaussian noise with a Rademacher vector ($\pm 1$) $u_i$ only changes the sign bits: $P^{(j)}$ is obtained by a bit-level sign flip of a previously computed vector, which degenerates into a single add / subtract operation. As a result, the kernel issues additions instead of a second matrix–vector multiply, making the update faster because additions are cheaper than multiplications on modern GPUs [1,2], as we briefly explained in Section 3.3.
>
> In practice, however, PyTorch currently does not provide low-level, optimized support for such Rademacher sign-flip kernels. To evaluate this effect on every large LLM would require re-implementing and integrating custom CUDA kernels for each model stack, which is a substantial engineering effort. As stated in the last paragraph of section 3.3, we implemented and profiled this optimization on a smaller model (OPT-125M). On OPT-125M with $N=8$, FZOO could implement a 1.92× speed-up over the “8perturbations + 8 forward passes” baseline.
>
>
> [1]Samuel Williams, Andrew Waterman, and David Patterson. Roofline: an insightful visual performance model for multicore architectures. Communications of the ACM, 52(4):65–76, 2009.
>
>
> [2] Hanting Chen, Yunhe Wang, Chunjing Xu, Boxin Shi, Chao Xu, Qi Tian, and Chang Xu. Addernet:Do we really need multiplications in deep learning? In Proceedings of the IEEE/CVF conference on computer vision and pattern recognition, pp. 1468–1477, 2020.

---

> ### Author Response · Authors · 2025-11-21
> **W2. Missing comparison with strong FO baselines (QLoRA / LoRA+Adam)**
>
> Thank you for the suggestion. First, we would like to clarify why we did not include strong first-order (FO) baselines such as LoRA+Adam in the main comparison. LoRA/QLoRA and prefix-tuning are important PEFT baselines, but these methods are **orthogonal** to our contribution: PEFT decides *which* parameters to update, whereas our work focuses on improving the **optimizer** for zeroth-order full-parameter tuning. FZOO can in fact be combined with PEFT (as we have already illustrated with prefix-tuning), so the two directions are complementary rather than competing.
>
> From a systems perspective, LoRA+Adam and QLoRA still require a backward pass and activation storage, so their **memory cost** remains higher than that of our ZO methods. In contrast, FZOO performs no backpropagation at all, keeping memory strictly at inference level. Given that our experiments already span multiple large models and 11 tasks, adding many FO–PEFT variants would significantly increase the experimental burden. For this reason, we focused the main tables on ZO methods that share the same inference-level memory footprint.
>
> We also agree with your point that adding strong FO baselines will make the contrast between ZO and FO clearer. Therefore, we plan to add Adam results to Table 2 (which reports performance) and Adam+LoRA results to Table 7 (which reports both performance and memory / throughput overhead).
>
> Running these additional experiments will take some time. Once the new results are ready, we will immediately submit an updated comment to supplement the rebuttal and will incorporate the corresponding Adam and Adam+LoRA rows into Tables 2 and 7 in the final version.
>
> The results in Table 2 with Adam added are as follows:
> | Model  | Method  | SST-2 |  RTE  |  CB  | BoolQ |  WSC |  WIC | MultiRC | COPA | ReCoRD | SQuAD | DROP | Average |
> |--------|---------|:-----:|:-----:|:----:|:-----:|:----:|:----:|:-------:|:----:|:------:|:-----:|:----:|:-------:|
> | Phi-2  | Adam    |   84.4  | 61.6 |    79.0  |   68.8    |  59.6  | 67.7  |  77.8  |  84.0 | 68.7 | 90.4 |  41.1  |   71.2 |
> | Phi-2  | MeZO    | 86.6  | 67.1  | 75.0 | 72.4  | 59.6 | 54.4 |  78.2   | 86.0 |  71.7  | 85.7  | 37.8 |  70.7   |
> | Phi-2  | HiZOO-L | 88.9  | 68.9  | 75.2 | 72.0  | 62.4 | 59.2 |  79.2   | 86.0 |  72.1  | 85.7  | 36.2 |  71.4   |
> | Phi-2  | FZOO    | 87.4  | 70.4  | 83.9 | 79.3  | 61.5 | 56.7 |  81.3   | 86.0 |  72.0  | 86.7  | 37.4 |  73.0   |
> | Llama3 | Adam    |  94.6  |  80.7  | 94.6 | 83.3 | 64.4 | 71.6 | 84.7  | 89 | 86.9 | 89.7 | - | - |
> | Llama3 | MeZO    | 92.2  | 74.4  | 69.9 | 76.7  | 63.5 | 57.8 |  77.6   | 88.0 |  85.6  | 86.7  | 57.1 |  75.4   |
> | Llama3 | HiZOO-L | 94.3  | 75.1  | 69.6 | 77.1  | 63.5 | 65.7 |  77.9   | 89.0 |  85.6  | 87.7  | 49.4 |  75.2   |
> | Llama3 | FZOO    | 94.3  | 77.6  | 69.6 | 81.8  | 65.4 | 60.8 |  81.5   | 88.0 |  85.3  | 87.9  | 56.5 |  77.2   |
> | OPT-13B| Adam    |  92.1 |  79.1  |  71.4 |  77.0 | 63.5 | 69.6 | 76.2 |  -  |  81.0 | 84.5 | -  |  -  |
> | OPT-13B| MeZO    | 91.4  | 66.1  | 66.0 | 67.6  | 63.5 | 59.4 |  57.3   | 88.0 |  81.7  | 84.7  | 30.9 |  68.8   |
> | OPT-13B| HiZOO-L | 92.1  | 68.2  | 67.9 | 66.5  | 62.4 | 69.4 |  61.1   | 89.0 |  81.1  | 63.6  | 22.7 |  67.0   |
> | OPT-13B| FZOO    | 93.7  | 71.1  | 69.6 | 72.2  | 63.5 | 60.5 |  66.0   | 87.0 |  81.0  | 84.8  | 28.7 |  70.7   |

---

> > ### Author Response · Authors · 2025-11-21
> > **Q2. How we get the reported 1.92× speedup?**
> >
> > Thank you for this question. Our thinking is as follows.
> >
> > 1. First, we would like to clarify that the speedup of FZOO can be decomposed into two **orthogonal** sources:
> >    (a) the **adaptive step size** schedule, and
> >    (b) an **efficient implementation of the batched forward pass**.
> >
> >    All speedup numbers reported in the figures of the paper come **only** from (a) adaptive step size. As we mentioned in W1, implementing (b) “efficient implementation of batched forward pass” on every large LLM would require re-implementing and integrating custom CUDA kernels for each model stack, which is a substantial engineering effort. We have not implemented this for every model, so we refer to the speedup after applying (b) as a **potential** speedup.
> >
> > 2. Because FZOO needs to compute the standard deviation, one optimization step involves multiple forward passes (e.g., $N = 8$). Compared with other two-point methods (e.g., MeZO, which is effectively $N = 2$), FZOO can benefit more from batch parallelism. Therefore, if we can optimize the forward pass from the batch perspective, FZOO can further amplify the effect of (a) adaptive step size. Based on this idea, we replace Gaussian noise with **Rademacher vectors**, so that many matrix–vector multiplications can be turned into additions.
> >
> > 3. Based on the above discussion, we can answer your question step by step:
> >    (1) Comparing against a **sequential FZOO baseline** (i.e., the sequential version of FZOO) is reasonable, because the reported 1.92× speedup is independent of the benefit from (a) adaptive step size. It is only intended to show that FZOO with multi-point perturbations and Rademacher vectors has **potential** to benefit from engineering optimizations.
> >    (2) Why not compare with realistic strong baselines like MeZO with $N = 1$? Here $N$ refers to the number of forward passes in one optimization step. For MeZO, $N = 2$. In addition, because MeZO uses Gaussian noise, it can benefit less from the kind of batched sign-flip implementation we use for Rademacher vectors.
> >    (3) Why not compare with two-sided ZO with efficient batching? For a two-sided ZO method, the minimal unit in one optimization step is two symmetric perturbations, and the next step can only start after the parameters are updated. Thus the batch size is relatively small and the benefit from batching is limited. If we group four such two-sided steps into one batch and combine them with Rademacher vectors (so that $N$ is aligned to 8), this would essentially be equivalent to our current sequential FZOO version. In that case, the extra time cost of computing the standard deviation in FZOO is negligible compared with the cost of the forward passes.

---

> ### Author Response · Authors · 2025-11-21
> **Q1. The definition of $d$ in Theorem 3.6**
>
> Thank you for raising this question. In all of our theoretical results, including Theorem 3.6, $d$ is **defined as the total parameter dimension**, i.e., the number of trainable parameters of the model. So $d$ should be understood as the usual “parameter dimension” in zeroth-order optimization, and the convergence rate in Theorem 3.6 explicitly depends on this $d$. The resulting bound is admittedly somewhat coarse (for example, the variance term scales polynomially with $d$), but this type of dependence on the parameter dimension is **standard** in random gradient-free methods such as Nesterov & Spokoiny’s random gradient-free minimization method [1], where the iteration complexity and variance bounds also scale with the ambient dimension $d$.
>
> We will make this definition explicit right before Theorem 3.6 in the revised version to avoid any ambiguity about the role of $d$.
>
> [1] Y. Nesterov and V. Spokoiny, *Random Gradient-Free Minimization of Convex Functions*, Foundations of Computational Mathematics, 17(2), 527–566, 2017.

---

### Official Review · Reviewer_1YPH · 2025-11-03

**Soundness:** 2
**Presentation:** 3
**Contribution:** 2
**Rating:** 4
**Confidence:** 4

**Summary:**

This paper introduces FZOO, a fast zeroth-order optimizer designed to reduce GPU memory in LLM fine-tuning. FZOO employs two primary strategies: it uses batched one-sided estimates to adapt its step-sizes based on loss variance and it utilizes Rademacher random-vector perturbations to accelerate per-batch computation. The authors demonstrate FZOO's effectiveness with experiments and provide theoretical convergence guarantees.

**Strengths:**

1. FZOO introduces a novel implementation that estimates more accurate zeroth-order gradients using batched one-sided estimates and Rademacher random-vector perturbations. This approach eliminates the two forward passes required by MeZO.
2. In experiments, FZOO demonstrates both improved speed-ups and higher model quality.
3. Similar to the original MeZO, FZOO is orthogonal to PEFT techniques.

**Weaknesses:**

1. I am unclear on the convergence results in Theorem 3.6 due to an apparent inconsistency in the definition of $d$ . The main text defines $d$ as the total number of model parameters; however, Appendix H. 1 clarifies that $d$ represents the per-layer input width, not the total parameter count. It is surprising that the zeroth-order method's convergence rate would be independent of the total number of model parameters.
2. The paper do not discuss and compare with an important, closely related line of work: hybrid first-order (FO) and zeroth-order (ZO) methods (e.g., [1,2]). A notable example, Addax [1], which adaptively combines FO and ZO gradient estimations based on input sequence length, is particularly relevant and warrants comparison.

[1] Li, Zeman, et al. "Addax: Utilizing zeroth-order gradients to improve memory efficiency and performance of sgd for fine-tuning language models." arXiv preprint arXiv:2410.06441 (2024).

[2] Chen, Jiahe, and Ziye Ma. "VAMO: Efficient Large-Scale Nonconvex Optimization via Adaptive Zeroth Order Variance Reduction." arXiv preprint arXiv:2505.13954 (2025).

**Questions:**

1. Could the authors provide results for Adam fine-tuning in their experiments to serve as a baseline comparison?
2. Would it be possible for the authors to include an experimental comparison with Addax [1]? Given that Addax is a prominent hybrid FO-ZO method, this comparison would provide valuable context for the proposed approach.
3. Could the authors provide a more detailed explanation of the term $d$ in their theoretical results? Specifically, clarification is needed on why  $d$ in Theorem 3.6 denotes the per-layer input width rather than the total number of model parameters, as the latter is a more common definition in this context.

I would be willing to reconsider my score if these concerns are satisfactorily addressed.

---

> ### Author Response · Authors · 2025-11-21
> **Thank you for the thoughtful and constructive comments**
>
> We sincerely thank you for the thoughtful comments and constructive suggestions, and in particular for carefully checking the consistency between the main text and Appendix. Your feedback has greatly helped us clarify both the theory and the empirical positioning of our work.

---

> ### Author Response · Authors · 2025-11-21
> **W1 & Q3. Inconsistency in the definition of $d$ in Theorem 3.6**
>
> Thank you for carefully spotting the inconsistency in the definition of $d$.
>
> In all of our theoretical results, including Theorem 3.6, $d$ is **intended to denote the total parameter dimension**, i.e., the number of trainable parameters of the model. Thus, the convergence rate *does* depend on the total number of parameters through $d$. The resulting bound is admittedly somewhat coarse, but this kind of $d$-dependence is standard in zeroth-order optimization [1], and our analysis follows the same spirit.
>
> The confusing wording in Appendix H.1 arose because we attempted to address a practical question: *how does the perturbation cost scale with $d$ in an actual implementation?* We wanted to explain that, since we generate perturbations layer-wise and only store a random seed per layer (similar to MeZO), the **extra cost for perturbations** grows with the maximum per-layer width rather than with the full parameter count $d$. However, the way this paragraph was written conflated this implementation-level quantity with the theoretical $d$ in Theorem 3.6, which led to the serious inconsistency you correctly identified.
>
> To avoid further confusion, in the revised version we will:
>
> - Explicitly state in the main text that in all theorems **$d$ denotes the total number of trainable parameters**.
> - Remove Appendix H.1 entirely, and instead add a brief remark in the main text that explains the implementation-level memory cost.
>
> ---
>
> [1] Y. Nesterov and V. Spokoiny, *Random Gradient-Free Minimization of Convex Functions*, Foundations of Computational Mathematics, 17(2), 527–566, 2017.

---

> ### Author Response · Authors · 2025-11-21
> **W2\&Q2. Compare with hybrid FO–ZO methods (Addax)**
>
> We appreciate the pointer to the hybrid first-order / zeroth-order line of work, especially the hybrid FO–ZO methods Addax (Li & Zeman et al., 2024) and VAMO (Chen & Ma, 2025). We agree these methods are relevant and will add a dedicated discussion in the final version.
>
> **Empirical strength of Addax.**
> After carefully studying Addax, we found that its design is **conceptually orthogonal to FZOO**, and at the same time its reported performance is substantially stronger than that of standard first-order baselines such as Adam, while FZOO remains a pure zeroth-order method. For example, on OPT-13B (numbers for Adam and Addax taken directly from the Addax paper, FZOO numbers from our own experiments):
>
> | Method      |   SST-2  |    RTE   |    CB    |   BoolQ  |    WSC   |    WIC   |  MultiRC |  ReCoRD  |   SQuAD  |
> | ----------- | :------: | :------: | :------: | :------: | :------: | :------: | :------: | :------: | :------: |
> | Adam        |   92.1   |   79.1   |   71.4   |   77.0   |   63.5   | **69.6** | **76.2** |   81.0   |   84.5   |
> | Addax       | **94.5** | **84.8** | **89.3** | **81.0** | **63.5** |   68.3   |   71.2   | **90.0** | **88.4** |
> | FZOO (ours) |   93.7   |   71.1   |   69.6   |   72.2   |   63.5   |   60.5   |   66.0   |   81.0   |   84.8   |
>
> Since the public Addax code currently lacks a complete environment.txt, we are still working on a full reproduction; the Adam and Addax numbers above are therefore taken from the original Addax paper.
>
> From these OPT-13B results we can see that FZOO **matches or exceeds Adam** on SST-2, CB, WSC, ReCoRD, and SQuAD, while Addax surpasses Adam on all tasks except WiC and MultiRC. The very strong performance of Addax is consistent with its design, which explicitly combines the strengths of first-order and zeroth-order methods.
>
> **Methodological relationship (orthogonality).**
> Conceptually, Addax and FZOO are orthogonal:
>
> - Addax is a **strategy for mixing FO and ZO from the data-batch perspective**: it routes short sequences in a batch to IP-SGD (FO) and long sequences to MeZO (ZO), without modifying the underlying FO or ZO optimizers themselves.
> - FZOO, in contrast, aims to **improve the ZO component itself** via normalization and structured perturbations, while keeping the memory footprint equal to inference and avoiding any backprop or gradient storage.
>
> Therefore, if FZOO outperforms MeZO under a given ZO budget, one can in principle build an **“IP-SGD + FZOO”** variant of Addax simply by replacing MeZO with FZOO in the long-sequence branch. **Once we manage to fully reproduce the Addax** codebase (their public repository currently lacks complete environment files, so this requires nontrivial engineering), we plan to implement this combination and  **submit new comment** to report the resulting gains.
>
> **Efficiency considerations.**
> From a systems perspective, Addax chooses ZO for long inputs and FO for short inputs. This tends to yield:
>
> - **Faster convergence**, thanks to the fast-convergence properties of FO updates on short sequences.
> - **Higher memory usage** than a purely ZO method, because under the same data batch a pure ZO optimizer is equivalent to replacing the FO part of Addax, and the IP-SGD branch still requires more memory than ZO due to activation / gradient storage.
>
> FZOO, on the other hand, does not attempt to optimize over the data batches: it uses ZO for both short and long sequences, so the training memory usage is **the same as inference-time memory**, but it cannot leverage FO gradients. Our design goal is precisely to **maximize ZO convergence speed under a fixed inference-level memory budget**. We will clarify this design philosophy and explicitly describe hybrid FO–ZO methods such as Addax and VAMO as complementary directions that could be combined with FZOO in future work.

---

> ### Author Response · Authors · 2025-11-21
> **Q1. Include Adam as an additional baseline.**
>
> Thank you for this helpful suggestion. We agree that including Adam baselines makes the performance gap between **pure ZO methods** and **standard FO fine-tuning** more explicit. We have already added **Adam rows to Table 2**, and the current results are shown below. For OPT-13B, the Adam row will use the numbers reported in the Addax paper.
>
> | Model  | Method  | SST-2 |  RTE  |  CB  | BoolQ |  WSC |  WIC | MultiRC | COPA | ReCoRD | SQuAD | DROP | Average |
> |--------|---------|:-----:|:-----:|:----:|:-----:|:----:|:----:|:-------:|:----:|:------:|:-----:|:----:|:-------:|
> | Phi-2  | Adam    |   84.4  | 61.6 |    79.0  |   68.8    |  59.6  | 67.7  |  77.8  |  84.0 | 68.7 | 90.4 |  41.1  |   71.2 |
> | Phi-2  | MeZO    | 86.6  | 67.1  | 75.0 | 72.4  | 59.6 | 54.4 |  78.2   | 86.0 |  71.7  | 85.7  | 37.8 |  70.7   |
> | Phi-2  | HiZOO-L | 88.9  | 68.9  | 75.2 | 72.0  | 62.4 | 59.2 |  79.2   | 86.0 |  72.1  | 85.7  | 36.2 |  71.4   |
> | Phi-2  | FZOO    | 87.4  | 70.4  | 83.9 | 79.3  | 61.5 | 56.7 |  81.3   | 86.0 |  72.0  | 86.7  | 37.4 |  73.0   |
> | Llama3 | Adam    |  94.6  |  80.7  | 94.6 | 83.3 | 64.4 | 71.6 | 84.7  | 89 | 86.9 | 89.7 | - | - |
> | Llama3 | MeZO    | 92.2  | 74.4  | 69.9 | 76.7  | 63.5 | 57.8 |  77.6   | 88.0 |  85.6  | 86.7  | 57.1 |  75.4   |
> | Llama3 | HiZOO-L | 94.3  | 75.1  | 69.6 | 77.1  | 63.5 | 65.7 |  77.9   | 89.0 |  85.6  | 87.7  | 49.4 |  75.2   |
> | Llama3 | FZOO    | 94.3  | 77.6  | 69.6 | 81.8  | 65.4 | 60.8 |  81.5   | 88.0 |  85.3  | 87.9  | 56.5 |  77.2   |
> | OPT-13B| Adam    |  92.1 |  79.1  |  71.4 |  77.0 | 63.5 | 69.6 | 76.2 |  -  |  81.0 | 84.5 | -  |  -  |
> | OPT-13B| MeZO    | 91.4  | 66.1  | 66.0 | 67.6  | 63.5 | 59.4 |  57.3   | 88.0 |  81.7  | 84.7  | 30.9 |  68.8   |
> | OPT-13B| HiZOO-L | 92.1  | 68.2  | 67.9 | 66.5  | 62.4 | 69.4 |  61.1   | 89.0 |  81.1  | 63.6  | 22.7 |  67.0   |
> | OPT-13B| FZOO    | 93.7  | 71.1  | 69.6 | 72.2  | 63.5 | 60.5 |  66.0   | 87.0 |  81.0  | 84.8  | 28.7 |  70.7   |
>
>
> In the revised version, we will:
>
> - Add **Adam rows to Table 2** for all models and tasks where full-parameter Adam fine-tuning fits into our 80GB A100 budget (e.g., phi-2, LLaMA-3-8B, and OPT-13B).
> - Clearly mark that **OPT-30B and OPT-66B** Adam fine-tuning is beyond our current compute and memory budget; for these very large models we focus on ZO methods precisely because FO training is prohibitively expensive.

---

> ### Comment · Reviewer_1YPH · 2025-11-24
> **Thank you for response**
>
> I am in generally satisfied with the authors' response and will raise my score to 6. Please ensure that the promised revisions (e.g., refining the theoretical analysis and including the new experiment results) are incorporated into the final manuscript. Thanks again for your detailed rebuttal.

---

> ### Author Response · Authors · 2025-11-24
> **Thank you for your response and support**
>
> Thank you very much for your follow-up and for your positive reassessment of our work. We also appreciate your reminder to ensure that the promised revisions are properly incorporated.
>
> After some effort, we have now identified a conda environment in which Addax can be reliably run, and we are in the process of conducting the corresponding Addax experiments. Although our compute resources are limited, we are confident that we can complete this part within the remaining rebuttal period. Once all experiments are finished, we will update the manuscript accordingly and revise the related comments to reflect the new results.
>
> I would also like to express my special thanks for drawing our attention to Addax. It has taught us a great deal: until now, our perspective on zeroth-order methods largely focused on the model side, and even when considering hybrid FO–ZO approaches, we tended to reason only from the parameter/optimizer viewpoint while underestimating the role of the data and batching scheme. In our future work, we will pay much closer attention to these data-side effects.
>
> If you have any further questions or suggestions, we would be very happy to discuss them.

---

> ### Author Response · Authors · 2025-11-26
> **Follow-up: Comparison with Addax**
>
> Following up on our previous response, we have successfully completed the full reproduction of Addax and conducted the promised experiments to verify the orthogonality between FZOO and hybrid FO-ZO methods.
>
> **Experimental Setup**
> Due to the time constraints and computational resource limits, we conducted these experiments on **OPT-2.7B**. During reproduction, we identified two distinct sets of hyperparameters in the official Addax repository:
> 1.  **Setting A (README):** $K_0=2, K_1=10, L_T=317, \alpha=1\text{e-}3$. This setting tends to bias results and memory efficiency towards IP-SGD (First-Order).
> 2.  **Setting B (Appendix):** $K_0=6, K_1=4, L_T=160, \alpha=5\text{e-}4$. This setting places more weight on the Zeroth-Order component.
>
> To ensure a comprehensive comparison, we evaluated both configurations. We implemented an **"Addax + FZOO"** variant by simply replacing the MeZO component in the long-sequence branch of Addax with our proposed FZOO, while keeping all other logic identical.
>
> **Results**
> The comparison results on 11 tasks are reported below (Average Accuracy). **Bold** indicates the best result within each comparison group.
>
> | Method | Hyperparams | SST-2 | RTE | CB | BoolQ | WSC | WIC | MultiRC | COPA | ReCoRD | SQuAD | DROP | **AVG** |
> | :--- | :--- | :--- | :--- | :--- | :--- | :--- | :--- | :--- | :--- | :--- | :--- | :--- | :--- |
> | **MeZO (Pure ZO FT)** | EPS=1e-3 | 92.2 | 58.8 | **62.5** | 64.0 | 53.8 | 54.2 | 58.4 | 76.0 | **75.0** | 77.6 | 25.3 | 63.44 |
> | **FZOO (Pure ZO FT)** | EPS=1e-4 | **93.6** | **69.0** | **62.5** | **70.8** | **57.7** | **57.5** | **65.5** | **83.0** | 74.8 | **81.8** | **26.5** | **67.52** |
> | | | | | | | | | | | | | | |
> | **Addax (MeZO)** | Setting A | **94.2** | **81.6** | 92.9 | 75.3 | 52.9 | 64.4 | 69.4 | 81.0 | 75.2 | 84.4 | 28.6 | 72.71 |
> | **Addax (FZOO)** | Setting A | 93.1 | 78.3 | **98.2** | **76.4** | **56.7** | **66.8** | **74.1** | 81.0 | **75.8** | **84.8** | **30.0** | **74.11** |
> | | | | | | | | | | | | | | |
> | **Addax (MeZO)** | Setting B | 92.2 | **79.8** | 92.9 | 63.1 | **61.5** | **61.6** | 44.5 | 82.0 | - | 81.5 | 24.6 | 68.37* |
> | **Addax (FZOO)** | Setting B | **93.6** | 76.5 | **96.4** | **75.7** | 59.6 | 57.4 | **61.4** | **84.0** | - | **82.3** | **24.6** | **71.16*** |
>
> *\*Note: For Setting B, the sequence length threshold ($L_T=160$) triggered runtime errors on tasks with longer inputs (e.g., "First Order Train dataset max length is too small"), necessitating the exclusion of ReCoRD. We also note that the original Addax paper similarly did not report results on ReCoRD and DROP.*
>
> **Analysis & Conclusion**
> The results clearly demonstrate that **FZOO consistently outperforms MeZO when integrated into the Addax framework**, regardless of the hyperparameter setting:
> * In **Setting A**, replacing MeZO with FZOO improves the average score from 72.71 to **74.11**.
> * In **Setting B**, the improvement is even more significant (from 68.37 to **71.16**).
>
> These findings empirically confirm our claim of **orthogonality**: FZOO serves as a more effective zeroth-order optimizer that can be used as a "drop-in" replacement for MeZO within hybrid FO-ZO strategies like Addax, further boosting their performance.
>
> Thank you again for suggesting this valuable comparison.

---

### Author Response · Authors · 2025-12-01
**Summary of Rebuttal Updates & Consensus (To Assist AC Assessment)**

**Dear Area Chair,**

To assist your assessment and reduce your workload following the recent system reset, we summarize below the key consensus reached with reviewers and the major experiments completed during the rebuttal period. **Please note that all quoted interactions occurred between Nov 24 and Nov 25 (UTC), strictly predating the public disclosure of the OpenReview incident (Nov 27).**

## 1. Explicit Score Increase (Reviewer 1YPH)
Reviewer 1YPH acknowledged our clarifications on the definition of $d$ and the hybrid method comparison.
* **Status:** **Consensus Reached.**
* **Evidence:** In the comment *"Official Comment by Reviewer 1YPH"* (Nov 24), the reviewer explicitly stated: **"I am generally satisfied... and will raise my score to 6."**
* **Action:** We have strictly followed through on the promised revisions, including the full reproduction of Addax (detailed below).

## 2. Concerns Resolved & Follow-up Answered (Reviewer ovaX)
Reviewer ovaX initially questioned theoretical positioning (clarified in **Appendix A.3 & B.4**) and later raised a follow-up on training stability.
* **Status:** **Theoretical Concern Resolved; Stability Follow-up Fully Answered.**
* **Evidence:** In the comment *"Official Comment by Reviewer ovaX"* (Nov 25 UTC), the reviewer **confirmed satisfaction with our theoretical clarification** (*"The primary clarification... is helpful"*). In the same comment, they raised a **new follow-up question** regarding the loss upturn in later training stages.
* **Action:** We immediately provided a comprehensive response analyzing the **loss dynamics** (identifying the cause as stochastic escape due to constant LR) and have added the supporting visualization to **Appendix H.5**. We believe this detailed analysis fully addresses the follow-up inquiry, although the review freeze prevented further confirmation from the reviewer.

## 3. Major Experimental Update: FZOO vs. Addax (Addressing 1YPH & jSd5)
Both Reviewer 1YPH and jSd5 requested a comparison with the hybrid method Addax. We completed a **full reproduction** during the rebuttal (results added to **Section 4.5, Table 8**).
* **Finding:** FZOO is orthogonal to hybrid methods. By replacing the ZO component in Addax (MeZO) with FZOO, we achieved strictly better performance on OPT-2.7B:
    * **Setting A:** Addax(MeZO) 72.71 $\rightarrow$ **Addax(FZOO) 74.11**
    * **Setting B:** Addax(MeZO) 68.37 $\rightarrow$ **Addax(FZOO) 71.16**
* **Conclusion:** This validates FZOO as a superior "plug-and-play" optimizer that enhances existing hybrid strategies.

## 4. Addressed Other Reviewers (pkyA & jSd5)
Although Reviewers pkyA and jSd5 have not yet replied prior to the freeze, we have fully addressed their requests in our submitted responses and revised manuscript:
* **For Reviewer pkyA:** We added **Adam baselines** to **Table 2** and clarified the speedup source as requested.
* **For Reviewer jSd5:** We completed the requested ablation studies (**Normalization vs. No-Normalization**, **Gaussian vs. Rademacher**) in **Appendix G.2** to quantify the gain from each component.

**Summary**
We believe the major concerns regarding theory (1YPH) and motivation (ovaX) were resolved, confirmed by an explicit acknowledgement of score raising. With the new comprehensive experiments (including Addax, Adam baselines, and detailed ablations) addressing the remaining suggestions, **we believe the revised manuscript is now robust and merits a positive recommendation.**

---

### Meta-Review · Area_Chair_KjZY · 2025-12-28

**Summary:**

This paper proposes FZOO, a zeroth-order (ZO) optimizer aimed at enabling full-parameter fine-tuning of large language models with inference-level memory while substantially improving convergence speed over prior ZO methods. The approach combines (i) batched one-sided ZO gradient estimation with variance-based normalization and (ii) Rademacher perturbations that facilitate batched forward passes and efficient implementation. Extensive experiments across multiple model scales and tasks show the effectiveness of FZOO.

Two reviewers (1YPH, pkyA) were generally positive and rated the paper at or above the acceptance threshold. Reviewer 1YPH’s initial concerns regarding theoretical notation and baseline clarity were carefully addressed in the rebuttal.

Reviewer ovaX raised more critical concerns regarding related-work completeness, the justification of variance-based normalization, and theoretical positioning. The authors responded with additional references, expanded discussion, and corrections to the analysis. While these responses may not fully eliminate all reservations, they substantially clarify the scope of the claims and place the method more clearly within the broader ZO optimization literature.

[Additional suggestion from AC] FZOO’s “split” between unperturbed activations and perturbed activations rhymes with the general idea of feature reuse / parallelization for ZO efficiency in DeepZero (Chen, A., Zhang, Y., Jia, J., Diffenderfer, J., Liu, J., Parasyris, K., Zhang, Y., Zhang, Z., Kailkhura, B. and Liu, S., 2023. Deepzero: Scaling up zeroth-order optimization for deep model training. arXiv preprint arXiv:2310.02025). This should be more clearly discussed in the submission.

Overall, the paper makes a meaningful and timely contribution to memory-efficient optimization for large language models. Despite some remaining limitations in related-work contextualization and end-to-end cost characterization, the technical soundness, empirical strength, and practical relevance of the approach justify acceptance. I therefore recommend acceptance, while encouraging the authors to further strengthen the related-work discussion and efficiency analysis in the final version.

**Reviewer Concerns:**

Addressed by the rebuttal:

Reviewer 1YPH’s concerns.

Reviewer pkyA’s questions on implementation details and speedup claims were partially addressed with clarifications on batching and the role of Rademacher perturbations.

Most of Reviewer ovaX’s concerns about the incomplete related-work discussion.


Still outstanding:

(Reviewer ovaX) The benefits and limitations of variance-based normalization are still justified mainly empirically, without a convincing broader analysis.

**Reviewer Scores:**

Reviewer 1YPH: Would increase the score to 6.

Reviewer pkyA: Likely would maintain the original score 6.

Reviewer ovaX: Likely would increase the score to 4.

Reviewer jSd5: Likely would maintain the original score 8.

---

### Decision · Program_Chairs · 2026-01-26

Accept (Poster)